# Evolving infectious disease dynamics shape school-based intervention effectiveness

Javier Perez-Saez [1,2,3,20] ✉, Mathilde Bellon[2,4,20], Justin Lessler [3,5,6], Julie Berthelot [1], Emma B. Hodcroft[7,8,9], Grégoire Michielin [10], Francesco Pennacchio [1], Julien Lamour[1], Florian Laubscher [11], Arnaud G. L'Huillier[11,12,13], Klara M. Posfay-Barbe[13], Sebastian J. Maerkl [10], Idris Guessous[14,15], Andrew S. Azman [2,3,16], Isabella Eckerle[2,4,21], Silvia Stringhini [1,15,17,21] & Elsa Lorthe [1,18,19,21] ✉ On behalf of the SEROCoV-Schools study group*

School-based interventions during epidemics are often controversial, as experienced during the COVID-19 pandemic, where reducing transmission had to be weighed against the adverse effects on young children. However, it remains unclear how the broader epidemiologic context influences the effectiveness of these interventions and when they should be implemented. Through integrated modeling of epidemiological and genetic data from a longitudinal school-based surveillance study of SARS-CoV-2 in 2021–2022 (N children = 336, N adults = 51) and scenario simulations, we show how transmission dynamics in schools changed markedly due to strong increases in community-acquired infections in successive periods of viral variants, ultimately undermining the potential impact of school-based interventions in reducing infection rates in the school-aged population. With pandemic preparedness in mind, this study advocates for a dynamic perspective on the role and importance of schools in infectious disease control, one that adapts to the evolving epidemiological landscape shaped by pathogen characteristics and evolution, shifting public health policies, and changes in human behavior.

Schools play an important role in the dynamics of human interaction and pathogen spread by facilitating interpersonal contact[1]. Schools have repeatedly been shown to be important amplifiers of respiratory pathogen transmission in the community for SARS-CoV-2, influenza and other diseases[2,3]. School-based non-pharmaceutical interventions (NPIs) to control transmission, such as limiting contacts between classes ("class cohorting"), are, however, particularly controversial due to their potential to disrupt the fundamental role schools play in child development and social cohesion[4]. The COVID-19 pandemic provided a stark example, with school closures impacting child mental health and exacerbating social inequities[5,6]. Hence, it is important that public health policy balances the utility of school-based NPIs for epidemic

control with its detrimental impacts on learning and childhood development[7,8].

Predicting the impact of school-based NPIs is limited by gaps in our understanding of respiratory disease transmission dynamics in school settings[9]. It can be difficult to conduct prospective studies in schools due to challenges in enrolling and maintaining participation of young children, their relatives and teachers[10,11]. Even studies based on population level surveillance data are hampered by difficulties in knowing exactly what policies are in place in various jurisdictions, as well as by inadequate measures of compliance and disease outcomes[12].

More fundamentally, the impact of NPIs can vary significantly over time due to changes in the epidemiological context. Pathogen

A full list of affiliations appears at the end of the paper. *A list of authors and their affiliations appears at the end of the paper. ✉e-mail: javier.perez@hug.ch; elsa.lorthe@hug.ch

evolution, socio-behavioral factors and community level interventions can all shape transmission dynamics, shifting the balance between within-school spread and community-acquired infections, thereby impacting the effectiveness of school-based NPIs[13]. Over the course of the COVID-19 pandemic, there were major changes in all these factors, yet their impact on transmission in schools and on school-based NPIs remains unclear.

Early evidence suggested that within-school infections caused by the initial SARS-CoV-2 strain were rare among young children (0–10 years), and that school-related attack rates mirrored those in the community[14]. Over time, as social distancing measures were relaxed and new SARS-CoV-2 variants that were more transmissible and evaded prior immunity emerged, reports of outbreaks in schools increased[15,16]. Serosurveys confirmed this increasing trend of infections in schools, showing that a large fraction of the world's school-aged population had been infected by the end of the first Omicron wave (April 2022)[17,18].

Changes in SARS-CoV-2 variants of concern (VOCs) and public health policies may, in part, explain why studies of school closure and school-based interventions have shown inconsistent results during the COVID-19 pandemic. Observational studies of transmission in schools have found varying levels of infection rates, lying on either side of those in the surrounding community[13,19]. Similarly, analyses of the impact of school closure on epidemic growth rates have shown a wide variation in effectiveness estimates across different countries and pandemic waves[7,20–22]. Our theoretical understanding of the reasons for these differences is hampered by the narrow scope of simulation studies, which mainly focused on the impact of interventions in the early phases of the COVID-19 pandemic[13,23–25]. The resulting incomplete understanding of the impacts of school-based interventions hinders our ability to draw lessons for future pandemic preparedness and response.

Here, we use data from the SEROCoV-Schools prospective school-based surveillance study in Geneva, Switzerland along with phylogenetic analysis and mathematical modeling to understand the relative contribution of school- and community-based transmission to SARS-CoV-2 incidence over a one year period, and explore the implications of these results on when and how to use school-based NPIs.

## Results

### School SARS-CoV-2 outbreaks became more frequent and larger in successive VOCs periods

The SEROCoV-Schools study enrolled 336 young children (age range: 1–7, 49% female), 51 educational staff (age range: 20–62, 91% female) and the households of participating children diagnosed with SARS-CoV-2 across 40 classes from two pre-schools and two primary schools in Geneva, Switzerland (Supplementary Table S1-S2, Supplementary Fig. S1). The baseline enrollment rate was 78% (268/343) for children and 96% (43/45) for staff across both school years (similar numbers in each year, Supplementary Tables S1-S2), with the remaining participants voluntarily joining during outbreak investigations. We conducted reactive outbreak investigations between March 2021 and February 2022, covering three successive periods of dominant SARS-CoV-2 variants (Alpha: March–June 2021, Delta: July-December 2021, and Omicron BA.1/BA.2: December 2021–February 2022, Fig. 1a), henceforth referred to as "VOC periods". Over the study period, community-level NPI measures were stable and of moderate stringency (Supplementary Fig. S2). They included 5-day quarantine for confirmed cases, requirement for a certificate of recent negative test or vaccination status for selected indoor activities, and limitations on large gatherings. Similarly, NPIs implemented by schools participating in the study were of moderate stringency and did not vary substantially during the study period. Further, SARS-CoV-V-2 RT-PCR testing was encouraged by public health authorities by reimbursing RT-PCR test costs for both symptomatic and asymptomatic individuals and, from 2021, by providing free rapid antigen detection tests (RADTs) each

month in pharmacies. Outbreak investigations were triggered by the report of any positive SARS-CoV-2 test result among either children or staff. Sampling included an oropharyngeal swab and capillary blood serology within 48 h of outbreak notification (D0), an oropharyngeal swab at five days after notification (D5) and serology at 30 days post notification (D30, Methods).

We investigated a total of 11 outbreaks that affected 20 of the 40 enrolled classes, from which 82.0% of children (401 of a total of 489 children-outbreaks) and 64% of staff (56 of 87 adult-outbreaks) provided at least one sample, with large variations in participation between outbreaks and classes (Supplementary Tables S3–S5). The frequency of outbreaks increased over time, coinciding with successive VOC periods (Fig. 1b, c), following a 10-fold increase in incidence in the general population (from ~150 confirmed cases per 100,000 per week during the Alpha and Delta periods to more than 2000 per 100,000 per week during Omicron, Fig. 1a).

Outbreak attack rates also increased in different VOC periods. As we were unable to measure these rates directly due to incomplete participation in most school classes, we developed a Bayesian framework to estimate attack rates by integrating data from viral detection (RT-PCR and RADT) and serology while accounting for missing data and imperfect test performance (Supplementary Material Section S1.8.1). This statistical model accounted for changes in the underlying individual-level probability of SARS-CoV-2 infection due to community-acquired infections or to within-school outbreaks which could vary between VOC periods and specific outbreaks. Consistent with previous reports[15,26], we estimate that outbreak attack rates ranged from 4% to 22% during Alpha, 3%–40% during Delta, and reached 25%–60% during the Omicron period (Supplementary Figs. S6–S7).

### Evidence of increasing frequency of viral introductions from the community

Contact tracing data and phylogenetic analyses suggest that the drivers of school-based outbreaks shift from intra-school transmission to community-based infection in different VOC periods. On contact tracing surveys, parents were increasingly likely to report that their infected child was exposed to a known COVID-19 case outside the school during the Delta to Omicron periods (insufficient responses during the Alpha outbreak; 74%, 32/43, during Delta; 82%, 28/34, during Omicron Supplementary Material section S1.3.3).

We gained additional insight into viral introduction frequency through phylogenetic analysis of 61 high quality school-based SARS-CoV-2 sequences from 7 of the 11 outbreaks covering all three VOC periods, supplemented by a large set of community sequences from Geneva and Switzerland (Supplementary Material Section S1.5, Supplementary Fig. S4). Phylogenetic results indicate a trend of increasing SARS-CoV-2 introductions from the community. The Alpha period outbreak, and one of two Delta period outbreaks, for which we have sequences, were likely driven by a single introduction, even though multiple introductions of the genetically identical virus cannot be ruled out (Fig. 2)[27]. In contrast, the second Delta period outbreak and all Omicron period outbreaks showed evidence of multiple introductions of the virus into schools from the community. These findings are consistent with phylogenetic studies in other school outbreak settings[11,28].

### Integrated evidence shows that rising community-acquired infections shaped within-school transmission dynamics

When considered in isolation, our contact tracing and phylogenetic evidence is limited by possible reporting biases, small sample size, and missing data. Here, we overcome these limitations by integrating all available epidemiological and genetic data using an individual-ased dynamic epidemic model and modern inference methods for coupled dynamical systems (Supplementary Material Section S1.8.2, Supplementary Figs. S8–S10). This dynamic model consisted of an SEIR-type

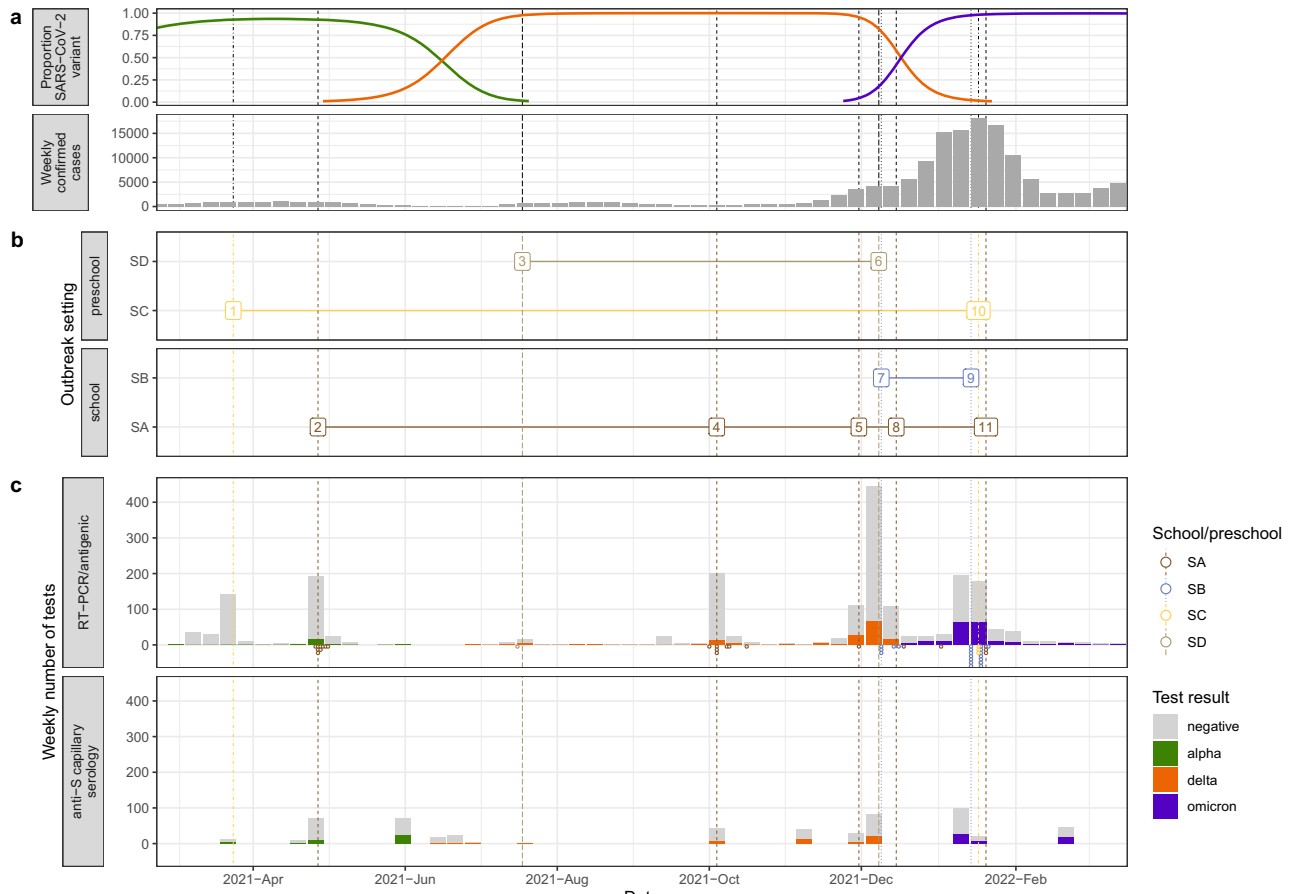

**Fig. 1 | Context and outcomes of the SEROCoV-Schools study. a** Study context in the state of Geneva (Switzerland) in terms of the proportion of main SARS-CoV-2 VOCs (top row) and the weekly number of reported positive SARS-CoV-2 cases (bottom row). **b** Study baseline/endline and SARS-CoV-2 outbreak detection dates (vertical lines indicate dates of first study visit) among the four prospected settings (two schools and two pre-schools), numbers represent the outbreak number. **c** Weekly number of RADT/RT-PCR tests and serologies from the study population.

Serologies were all collected within the study protocol at baseline, endline and during outbreaks at Day 0–2 and Day 30 (Methods). RADT/RT-PCR tests were either collected at baseline, during outbreak investigations (at Day 0–2 and Day 5–7), or tests taken by study participants outside of the study and either reported to our team through questionnaires or notified to Geneva's Directorate of Health centralized SARS-CoV-2 test result repository. The number of SARS-CoV-2 sequences analyzed are indicated by vertical stacked dots by date of collection.

model that simulates SARS-CoV-2 transmission events between individuals in schools, accounting for contact networks between students and staff (within- vs. between-group transmission), VOC-specific natural history parameters (incubation period and generation time) and levels of cross-variant immunity following infection or vaccination. This integrated model-based analysis allows us to reconstruct the underlying infection patterns in our study population and estimate how the parameters governing school-based transmission and infection pressure from the community changed between VOC periods.

Our main inference is that, during each new VOC period, the probability of community-acquired infection increased by about an order of magnitude, while individual-level within-school transmission rates increased only slightly (Fig. 3a, b). We estimated that the weekly probability of infection from the community was less than 0.1% during the Alpha period (Maximum Likelihood Estimate, MLE: 0.08%, 95% confidence interval, CI: 0.07–0.1), increasing to 0.5% during the Delta period (95% CI: 0.3–0.7), and up to more than 15% during the Omicron BA.1 period (MLE: 16%, 95% CI: 12–20). Across VOC periods, the probability of being infected by a single infectious student in the same class was close to 5% per infectious period (~5 days). The probability of infection by a single infectious student in another class was 0.3% during the Alpha and Delta periods, and 2% during the Omicron period. These estimates match those of epidemiological studies investigating outbreaks across different variants in school settings[29,30].

As a result of increasing community infectious pressure, any given infection during outbreaks in our study schools was more likely to have been imported from the community during the Omicron period (30% probability, interquartile range across individuals, IQR: 19–35) compared to the Delta (18% probability, IQR: 7–27) and Alpha periods (0.03% probability, IQR: 0.02–0.1), consistent with our phylogenetic analyses. Higher community-infectious pressure fueled subsequent within-school transmission, leading to an increase in the overall probability of infection with each new VOC period, going from peaks at less than 1% probability of infection per day during Alpha (range of peaks: 0.1%–0.8%), to around 2% during Delta (range of peaks: 1.2%–2.6%), and reaching more than 10% during Omicron (range of peaks: 13%–18%, Fig. 3c).

Our results further show that SARS-CoV-2 transmission in schools did not in fact mirror infection rates in the community, and the relationship between incidence in the community and incidence in school children changed between VOC periods. We estimated true community-level infection rates by correcting weekly SARS-CoV-2 confirmed cases reported by the Republic and Canton of Geneva Directorate of Health[31], with VOC-specific case-to-infection ratios derived from our repeated cross-sectional seroprevalence surveys[32–34]. Comparing these estimates with model-reconstructed incidence rates in schools, we find that school-based infection rates were generally lower than those in the community during the Alpha period (Fig. 3d),

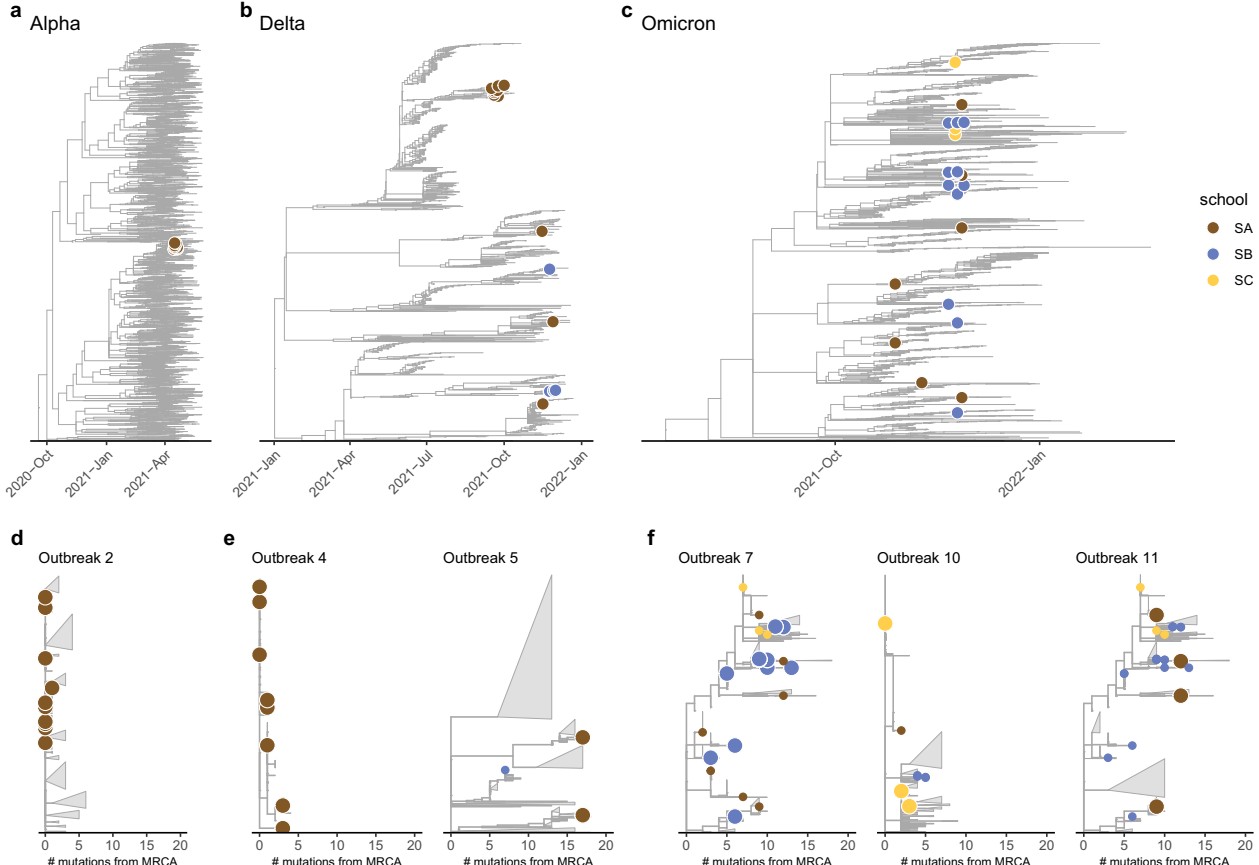

**Fig. 2 | Phylogenetic trees of study sequences identified among local, national and international sequences. a–c** Study participant sequences (dots, color indicates study school, no sequences available from school SD) shown within the inferred time-scaled phylogenetic trees including contextual sequences from Geneva, the rest of Switzerland and internationally (Methods), cut to the most recent common ancestor (MRCA) of the three dominant SARS-CoV-2 VOCs circulating in Geneva during the study period for the Alpha (**a**), Delta (**b**) and Omicron

BA.1 (**c**) periods (Fig. 1a). Mutation-scaled trees by outbreak in terms of number of mutations from the MRCA of study sequences (large dots indicate study sequences in outbreak, small dots study sequences in other schools), grouped by dominant SARS-CoV-2 variant: Alpha (**d**, Outbreak #2), Delta (**e**, Outbreaks #4 and 5), and Omicron BA.1 (**f**, Outbreaks 7, 10 and 11). Clades that do not contain study sequences were masked as gray triangles.

but school incidence rates were on average two to five times higher than those in the general population during the Delta and Omicron periods.

## School-based interventions can fail when the probability of community-acquired infections is high

These results show a shifting relationship between school- and community-based transmission dynamics over subsequent VOC periods. These shifting relationships may have profound impacts on school-based NPI effectiveness, raising the question of when such measures should best be deployed in future outbreaks of respiratory pathogens.

We performed scenario simulations of synthetic school-based NPIs leveraging our parameterized SARS-CoV-2 transmission model. We took the Alpha, Delta and Omicron periods as exemplars of epidemiological contexts along a gradient of increasing transmission potential and community infectious pressure, noting that these contexts are driven both by difference in pathogen characteristics as well as contextual socio-behavioral factors. Synthetic interventions were designed to account for the overall effect of NPIs as percent reductions in within- and between-class transmission rates. Hence, these simulation scenarii represent the combined effect of typical school-based NPIs such as masking, class cohorting, air purification, or quarantine measures, but do not attempt to directly model the impact of any one

specific intervention[35,36]. For each transmission setting (Alpha-, Delta- and Omicron-like periods), we performed 1000 simulations of 3 month-long transmission dynamics for each pair of reductions in within- and between-class transmission rates, ranging from 0% (no change, i.e. baseline conditions) to 100% reduction (complete interruption of transmission). The scenario of 100% reduction in both within- and between-class transmission corresponds to school closure, where infections can only be acquired in the community.

Simulations show that infection pressure from the community is a key constraint on the effectiveness of NPIs in controlling school-based respiratory disease epidemics. This is because increased infection in the community increases both the number of students infected outside of school and the opportunities for onward transmission in school due to these introductions. Specifically, we found a strong relationship between increased school-based NPI stringency and reduced incidence in schools in simulations of Alpha-like (very low community infectious pressure) and Delta-like (low community infectious pressure) contexts, with minor differences between pre-school and primary school settings (Fig. 4). In contrast, school-based interventions failed to reduce cumulative infections among students in the Omicron-like scenario (high community infectious pressure). Even in scenarii corresponding to school-closure (i.e., no within-class nor between-class transmission), only a median of 9.5% of infections were averted in these high transmission settings. These results were robust to different

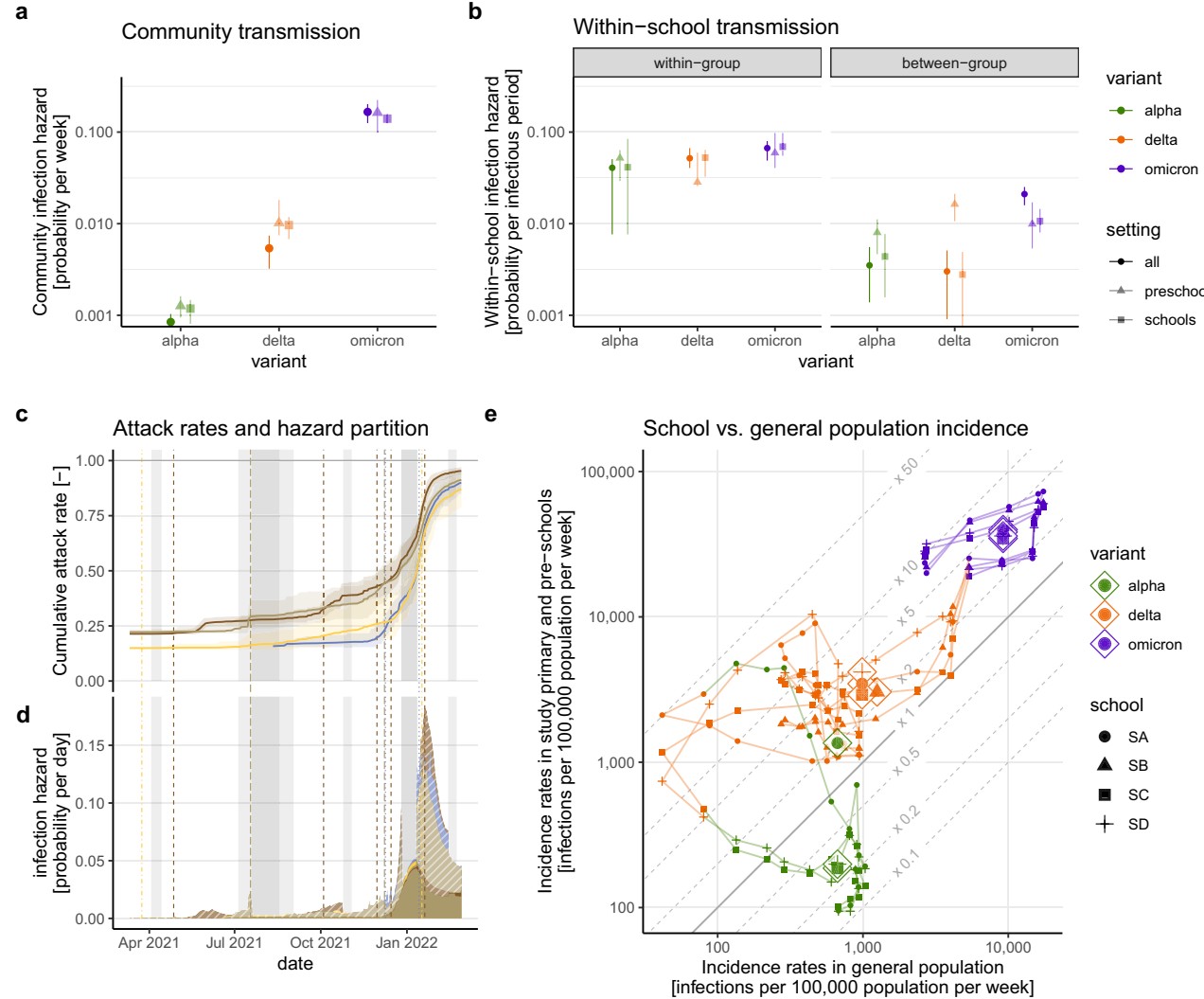

**Fig. 3 | Mathematical modeling inference of changes in school-based SARS-CoV-2 transmission dynamics with changes in dominant VOCs. a** Estimated parameters of the probability of infection per week for community transmission (dots give the maximum likelihood estimate, error bars the 95% Monte Carlo profile CIs), with overall estimates and stratifying by educational setting (primary schools vs. pre-schools, colors). **b** Same as (**a**) for within-school transmission parameters, differentiating within-class and between-class transmission in terms of the infection probability per single infectious period. **c** Inferred cumulative SARS-CoV-2 attack rate in children, (lines give the mean value across trajectories of 500 smoothing distribution draws and children, shaded areas give the 95% quantile interval).

**d** Inferred mean total daily infection hazard by school and its partition between community-acquired transmission (full areas with dark colors), and within-school transmission (light color, hashed areas). Colors indicate schools following Fig. 1. **e** Comparison of within-school and general population SARS-CoV-2 incidence rates. Lines show school trajectories in time, small dots indicate weeks by dominant VOC period (point types). Large circle dots indicate trajectory centroids (mean over the period of school and general population incidence) by variant and primary or pre-school. General population incidence rates were computed based on reported positive cases corrected for infection under-reporting.

durations of epidemic simulations (6 and 12 months) (Supplementary Figs. S12, S13), and when accounting for feedback between reductions in within-school and community-wide transmission (Supplementary Figs. S12, S14).

## Discussion

Through an integrated analysis of high-resolution epidemiologic and virological data, this work reveals the degree to which within-school pathogen transmission dynamics can change in time as a function of the level of community-acquired infections, here exemplified with the COVID-19 pandemic, and its potential impact on school-based NPI effectiveness. Scenario simulations demonstrate that school-based interventions can be highly effective when community transmission is low and thus community-acquired infections by school-aged children rare. However, when community transmission rates are high, school-

based interventions can only be implemented effectively as part of a comprehensive suite of interventions that target transmission in multiple settings. This contrasts with how school-based NPIs tended to have been implemented, especially during the VOC-dominated phase of the COVID-19 pandemic (Supplementary Material Fig. S15), where high community transmission typically triggered mitigation measures up to including school closures[20,21,37]. While there may be valid reasons for maintaining school-based interventions during periods of high epidemic activity, such as protecting vulnerable students or addressing staffing shortages, epidemic control in terms of reductions of infections in school-aged populations may not be within practical reach. Rather, school-based interventions are most impactful when incidence is low but epidemic potential remains high, as they can reduce the risk of disease among students and may help prevent larger community outbreaks.

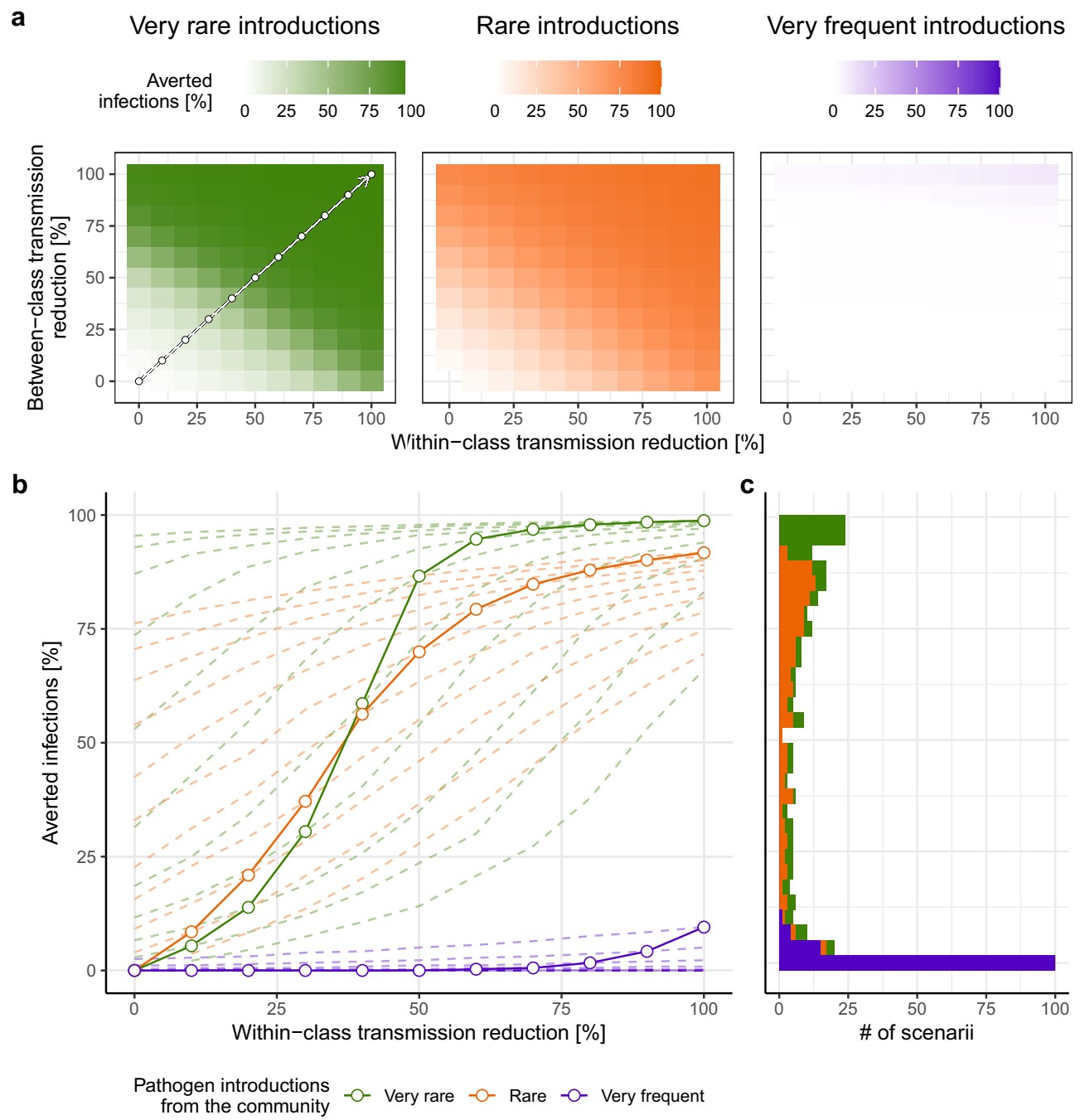

**Fig. 4 | Scenario modeling simulations of schools-based NPI effectiveness under different epidemiological contexts and intervention levels.** Proportion of infections averted in school population in terms of the reduction in final attack rates (median of 1000 simulations) as a function of within- and between-class transmission reduction across varying epidemiological contexts ranging from very rare pathogen introductions from the community representing the Alpha SARS-CoV-2 period as inferred in this study (~0.1% probability per person per week, green, **a**), to rare introductions (Delta-like setting ~1% probability per person per week, orange), and very frequent introductions (Omicron-like setting, ~ 15% probability per person per week, purple). The arrow in the first subpanel represents parameter combinations illustrated by full lines in (**b**) (diagonal line, equal within- and between-class reductions). **b** Proportion of infections averted in schools as a function of increasing within-class and between-class transmission reduction (full lines and dots), or across constant levels of between-class transmission reduction (dotted lines). **c** Histogram of infection reduction across scenarii by epidemiological context.

This potentially counter-intuitive relationship poses challenges for using surveillance data and predictive modeling to inform effective policy decisions. In particular, determining when to implement school closure or extensive school-based NPIs requires accurately identifying periods of high epidemic potential but low incidence, and effectively communicating these findings in a way that gains the trust of both policy makers and the community. Equally challenging is

understanding when additional school-based measures are unlikely to have significant impact, meaning that their negative consequences likely outweigh their public health benefits, allowing a shift toward "normal" school activity, even in the midst of a widespread community epidemic. In these situations, preventing infections in school-aged populations therefore requires effective transmission control in the community, rather than school-based interventions. Identifying these

situations will benefit from continued efforts in real-time inferential modeling of changes in epidemic progression[38], as well as in strategic modeling to compare alternative scenarios of epidemic drivers and public health interventions[39].

Taken together, these conclusions highlight the importance of continuous epidemiological surveillance to track community-level infection rates, the emergence of novel pathogen variants, and changes in socio-behavioral factors favoring transmission when assessing the adequacy of different NPIs. In fact, the level of epidemiologic and genetic data granularity attained in our study comes at the price of a small number of investigated schools and may be impractical at larger scales. More generally, our conclusions on the sensitivity of NPI effectiveness as a function of community-wide transmission intensity may extend to other types of epidemic control interventions such as contact tracing. Beyond school-based interventions, NPIs are typically considered in packages which may further modify the relation between epidemiologic contexts and the effectiveness of single NPIs, thus highlighting the importance of relevant population-level epidemic indicators to inform public health decision making. Determining which indicators should be monitored to identify "NPI tipping points" such as the one highlighted in this study should be the matter of future research for pandemic preparedness, especially given these are likely to vary across pathogens and epidemics. It seems likely though that these indicators will rely on robust surveillance through classical case reporting and novel methods such as wastewater monitoring[40], as well as enhanced genetic monitoring through routine and large-scale sequencing to identify emerging pathogen variants and track their spread[41]. In addition, enhanced epidemiologic surveillance may also benefit from an integrated assessment of socio-behavioral and policy factors that have the potential to impact community-level transmission patterns, as exemplified by the strong changes in human mobility patterns during the COVID-19 pandemic[42].

This study benefits from incorporating phylogenetic, community and individual level epidemiologic information through an integrated modeling framework, but it has limitations. It focuses on only four education settings within a single geographic area, and variability in participation rates across outbreaks limited direct epidemiological interpretation. We overcame data missingness through mathematical modeling, which we anchored in existing estimates of key natural history parameters available for the VOCs in this study. Our approach involves assumptions, such as equal infectivity between adults and children across SARS-CoV-2 variants based on evidence from the ancestral strain[43,44], and homogenous mixing within classes, constant community infectious pressure and within-school transmission parameters during VOC periods. This masks potential changes in COVID-19 management in schools not related to isolation and quarantine duration imposed by health authorities, which we have accounted for. As such, parameter estimates in each VOC period should be interpreted as integrating across SARS-CoV-2 natural infectivity and contextual factors affecting transmission. Although changes in inferred transmission parameters cannot fully be attributed to VOCs alone, we note that during the study period NPI stringency in Switzerland was relatively stable (Supplementary Fig. S2), and that measures of human mobility did not show strong variations (Supplementary Fig. S16). In contrast, the emergence of the Alpha, Delta and Omicron subvariants has been consistently associated to strong increases in incidence globally, in particular for Omicron[18], thus highlighting the intrinsic potential of pathogen variants to shape epidemiologic patterns and motivating the focus in this work on the distinct periods of shifting VOC dominance. Our inferential estimates on within-school transmission might be influenced by the indirect effect of our study on school control measure implementation and adherence, which we cannot control for. It is however unlikely that this eventual indirect effect is large enough to undermine our main conclusions on the relative importance of within-school vs. community-acquired transmission.

The consistency of estimates between pre-schools and primary schools is reassuring.

Finally, the results and conclusions of our scenario simulation investigations on the use of school-based measures in varying transmission settings are anchored in our study of SARS-CoV-2, and the same type of comparison may yield different quantitative results using natural history parameters and age-specific transmission rates for other respiratory pathogens, for instance due to age-specific immunity profiles exemplified by the 2009 influenza pandemic[45]. We note that our simulations focused on infection rates in the school-aged population, and not in resulting community-level changes in transmission intensity which would have required a full simulation of community-level transmission dynamics. Because our aim was to provide a qualitative view on the potential impacts of changing levels of community transmission on school-based NPIs, we considered reduction in school-aged population infections a sufficient proxy of population-level impact as it maps to subsequent reduction in secondary transmission events, echoing studies with similar aims[13]. Future modeling studies aimed to, and tooled for, assessing the effect of school-based NPIs on community-wide epidemic patterns may leverage these insights in the assumptions on how NPI stringency is mapped to the effective contribution of schools to onward community transmission. We nevertheless expect that our main qualitative result, i.e. that school-based interventions loose effectiveness when the probability of community-acquired infection is too high, is of policy relevance across a wide space of respiratory pathogen characteristics and may be broadly relevant to infectious diseases involving school-aged children. The clear definition of what "too high" is remains subject of future work for specific pathogens and transmission settings of public health importance, in particular to assess the sensitivity of losses in school-based NPI effectiveness to age-specific profiles in pathogen exposure, susceptibility and infectivity.

School-based interventions during the COVID-19 pandemic have been particularly controversial, due to tensions in balancing transmission reduction with adverse impacts on children. This study adds nuance to these debates by highlighting the pitfalls of taking a static view on the role and importance of schools for disease control. It shows the need for a dynamic view that accounts for the evolving epidemiological context. This dynamic approach reveals that optimal policy may sometimes be counter-intuitive, hence highlighting a need for creating trust and clear communication channels between communities, policymakers and the epidemiologic community. Building these trusted relationships will be essential if we are to effectively respond to future pandemic threats.

## Methods
### Study design and setting
SEROCoV-Schools is a longitudinal, prospective, class-based surveillance study, which aims to describe the transmission dynamics of SARS-CoV-2 in primary schools and pre-schools and the risk of introduction of the virus into the children's households. A convenience sample of two primary schools and three pre-schools in the canton of Geneva was included. All classes and groups of children in the target age group (1–2 to 6–7 years old) were eligible. Parents and teachers were informed about the study orally and through a flyer and a detailed information note. Children received an age-appropriate information letter about the project and a schema explaining the process of infection and antibody formation. All those who consented to participate were included, with participation rates at baseline ranging from 33% to 100% across classes.

A baseline visit was organized for all the included pupils, teachers, and school staff members at the beginning of the study in March 2021 and for new participants at the beginning of the second school year in September 2021. The assessment consisted of a SARS-CoV-2 serology on capillary blood, a SARS-CoV-2 antigen rapid diagnostic test, or a

SARS-CoV-2 reverse transcription polymerase chain reaction on oro-pharyngeal swabs and the completion of an online questionnaire on the Specchio-COVID19 secured online platform[46] (including questions about socio-demographic characteristics, household composition, health condition, previous SARS-CoV-2 infection and SARS-CoV-2 vaccination status). Teachers also filled in a questionnaire about the school/pre-school organization and the measures implemented in their institution to prevent SARS-CoV-2 transmission.

During the surveillance phase (from March 2021 to February 2022), the participants or their parents were invited to report any contacts with infected people, symptoms compatible with a SARS-CoV-2 infection or positive SARS-CoV-2 tests. An outbreak investigation was triggered in a class when a participant (child or adult) tested positive. The date of this first positive test was defined as Day 0. The investigation consisted of three successive visits in the class: (1) between Day 0 and Day 2 with serology and RADT or RT-PCR test, (2) between Day 5 and Day 7 with RADT or RT-PCR test, and (3) on Day 30 with serology. After each visit, the participants or their parents completed a questionnaire on COVID-like symptoms, contact with positive cases, and additional SARS-CoV-2 tests. An investigation including the same three visits was conducted in the households of all positive children whose family members consented to participate. At the end of the two school years (i.e., in June 2021 and June 2022), all participants were offered a final serology.

This study was approved by the ethics committee of the Canton of Geneva (Project ID 2020-02957). All parents and teachers gave written consent, and the children gave verbal assent to participate.

## Study population

The present study population consisted of all participating children and teachers (including assistants). In case of an outbreak, household members of infected pupils were also invited to participate (defined as people living with the positive child at the time of their positive test). Non-participating pupils or non-teaching staff (administrative, cleaners, catering) members of the school/pre-school staff were also invited to participate in the outbreak investigations or to get tested outside of the study and to report the test result to our team. Therefore, the number of participants varies between the baseline assessment and each outbreak investigation (Supplementary Fig. S1).

We obtained study participation consent from 336 children and 51 educational staff (children: 202 in term 2020/2021 and 268 in term 2021/2022; adults: 30 in 2020/2021 and 43 in 2021/2022). Of these, baseline information was provided by 181/202 and 217/268 children and 30/30 and 34/43 adults in terms 2020/2021 and 2021/2022 respectively. Participants came from 44 distinct classes across two primary schools and three pre-schools (20 classes in term 2020/2021 and 24 in term 2021/2022).

We excluded from the analysis presented here one pre-school with only a baseline visit and a total of 22 adults and 18 children who consented to participate. Several positive cases were identified at the end of January 2022 and at the beginning of February 2022, but no outbreak investigation could be carried out for logistical reasons. Longitudinal data is presented in Supplementary Fig. S3.

## Epidemiological investigations

**Case definition.** A confirmed SARS-CoV-2 positive case was defined by a positive RADT and/or RT-PCR, and/or a seroconversion between Day 0–2 and Day 30 (from seronegative to seropositive according to the test-specific cut-off, unrelated to vaccination). Confirmed cases could be further classified as symptomatic or asymptomatic.

**Other variables.** The total number of pupils and teachers in the classes or groups included was indicated with the variable "N total". In households, "N total" represents the number of people living in the household of a SARS-CoV-2-positive child. An individual was considered a participant if they had signed an informed consent form ("n participant"). A participant was considered to have been investigated in an outbreak ("n investigated") if they had had at least one RT-PCR or RADT within 21 days of identification of the index case and/or at least one serology during the outbreak visits. Only data from participants (with a signed consent form) were presented in the descriptive table (Supplementary Tables S1–S5). Participants under 18 years of age were considered children.

Self-reported or parent-reported variables included sex, age (on the first day of the baseline school visit for the baseline table and on the day of the first outbreak visit for the outbreak table), perceived health status, presence of any chronic disease, SARS-CoV-2 vaccination status (considered positive with at least one dose of vaccine), and presence of symptoms compatible with COVID-19 (reported only for SARS-CoV-2 infected participants). Baseline vaccination status was not reported for children, as vaccination started in January 2022 for this age group. The number of participants eligible for vaccination was determined according to the age of the participants and the age groups for which vaccination was available in Geneva at the time of the outbreak[47]. We did not consider the presence of vulnerable participants in the number of participants eligible for vaccination for whom vaccination could have been available earlier.

Test results were considered as categorial variables: positive/negative for RT-PCR and RADT, positive/undetermined/negative for serologies according to the specific threshold for each test.

**Contact-tracing.** Contact-tracing interviews were performed with all positive participants or their parents using a structured questionnaire about symptoms, contact with infected people, school attendance and out-of-school activities and gatherings in the 14 days before diagnosis.

**Virological testing (RADT and RT-PCR).** To detect a current SARS-CoV-2 infection, the Panbio COVID-19 Ag Rapid Test from Abbott was used from March to September 2021. This test was performed using oropharyngeal swabs as this sampling method has been validated in adults and is less invasive than nasopharyngeal swabs and, therefore, more acceptable to young children[48]. The results were confirmed by an RT-PCR on the swab leftovers (in-house SARS-CoV-2 RT-qPCR as previously described[49]). Due to the limited sensitivity of the RADT in children reported in the literature and observed in our study[15,50], it was replaced in September 2021 by an RT-PCR (Cobas® SARS-CoV-2 Test, Cobas 6800, Roche) in oropharyngeal swabs. All results were communicated to the participants as soon as they became available. When participants took tests outside the study, the tests and collection methods varied. The results of these tests were also entered into the database when the participants communicated them to our team. Whenever possible, positive samples were obtained from the labs, re-tested with RT-PCR (Cobas® SARS-CoV-2 Test), and analyzed by sequencing like the other study samples.

**Sequencing and phylogenetic analysis.** SARS-CoV-2 whole genome sequencing was performed for positive samples at the Health 2030 Genome Center (Geneva) using the Illumina COVIDSeq library preparation reagents following the protocol provided by the supplier. Only the sequences with coverage higher than 90% were included in the present analysis (at least 27,000 bases). The consensus sequences were verified one by one by a bio-informatician and then submitted on GISAID (sequence list in Supplementary Fig. S2).

Details on analyzed sequences and phylogenetic inference are given in the supplementary material.

**Serological testing.** The serological tests consisted of an anti-SARS-CoV-2 Spike IgG high-throughput microfluidic nano-immunoassay performed on dried whole capillary blood from a finger pick collected with a microsampling device (Mitra® Clamshell, Neoteryx, USA) as

previously described[51,52]. This sampling method has shown good performances in adults, is minimally invasive, easily implementable in schools and more acceptable than traditional venipuncture by children. Serologies were communicated to the families approximately 2 months after the visit.

**Descriptive analysis.** Descriptive analyses (Supplemental Tables S1–S5) were performed with the software Stata 17.0. Quantitative variables are described with median, interquartile range, and minimal/maximal values. Categorial variables are described with ratios and percentages.

### Modeling analyses

**Overview.** This study had two main goals: (i) to infer the the relative contribution of within-school transmission vs. community importation of SARS-CoV-2 in the study population and how these changed in time (inferential modeling goal), (ii) and building on these inferential results, to investigate the implications of changing strength of community importations on the effectiveness of school-based interventions against respiratory pathogens (strategic/scenario modeling goal). We addressed our study goals by developing two modeling frameworks: a statistical model and a dynamic transmission model. Full model details are given in the Supplementary Material Section S1.8.

We leveraged both the statistical and dynamic modeling frameworks for the inferential modeling goal because of their complementarity in the targets of inference they were equipped to pursue, and to strengthen the robustness of our results. The common target of inference of both modeling frameworks were (i) the cumulate attack rate during SARS-CoV-2 outbreaks in our study schools (which could not be evaluated directly because of data missingness), and (ii) the community-imported transmission rate and how it changed with SARS-CoV-2 VOC periods. A key target of inference of the statistical model was the time-varying sensitivity of the capillary-blood serological assay we used in this study, which had not been previously evaluated in children and with time post-infection. These estimates of time-varying sensitivity were in turn an input to evaluate the likelihood function of the dynamic model. By explicitly representing SARS-CoV-2 transmission events, the dynamic model allowed to refine inference on within-school vs. community by incorporating the genomic data produced in this study, and further differentiating between within-class and between-class transmission.

The dynamic modeling framework allowed to pursue the strategic/scenario modeling goal on the effectiveness of school-based interventions by explicitly manipulating the within-class and between-class transmission rates. Using inferred parameters as baselines, we investigated the effect of reductions in within-school transmission rates on overall attack rates under different scenarios of community-acquired infections.

### Statistical modeling framework

**Model description.** The aim of the statistical modeling framework was to infer the time-varying sensitivity of capillary blood serology in our study population as well as to infer outbreak cumulative attack rates by linking time changes in school-group level seroprevalence to serological and virological testing data. A full description of the model is given in Supplementary Material Section S1.8.1.

The statistical model tracks the individual-level probability of SARS-CoV-2 infection in time intervals characterized by constant infection hazard rate. We account both for the probability of infection from the community assuming a constant hazard rate within each time interval, as well as within-school infection during periods where we observed outbreaks.

**Inference.** We link individual-level trajectories of SARS-CoV-2 probability of infection to capillary-blood serology results as well as

virological testing data. The sensitivity of the serological assay had not been evaluated in children and as a function of time post-infection, we thus here infer time-varying sensitivity using cubic regression following previous studies[53]. The model further accounts for unknown dates of infection in the absence of positive virological test data, and incorporates prior information on SARS-CoV-2 seroprevalence in the general school-aged population based in the state of Geneva[33,34]. Inference was drawn in a Bayesian framework using Hamiltonian Monte Carlo as implemented in the Stan programming language[54]. Priors and algorithmic parameters given in Supplementary Materials Section S1.8.1.3.

### Dynamic modeling framework

**Model description.** We model the transmission of SARS-CoV-2 in our school population using a stochastic SEIS-type compartmental model at the individual level that accounts for social contact networks and community-acquired infections. We leverage the dynamic model in two ways, first for inferring the relative importance of within-school vs. community infections, and then to simulate and evaluate scenarios of school-based interventions. A full description of the model is given in Supplementary Materials Section S1.8.2.

Each modeled individual can be in one of four states (susceptible [S], exposed [E], infected [I], susceptible with partial protection from previous infection [S']). Following previous modeling studies of school-based epidemics[1], we explicitly account for contact networks to model transmission events. In the model contacts can be one of two types, either within-class or between-class, and each have a different transmission rate parameters. We further account for community-acquired infections through a hazard rate that changes with each period of SARS-CoV-2 VOC dominance (Alpha, Delta, and Omicron, Fig. 1). We use published estimates to have natural history parameters be VOC-specific (incubation period, generation time)[55,56], as well as to account for partial immunity following cross-variant infection and vaccination[57].

**Inference.** Our main targets of inference were the within-class and between-class transmission rates, and the community-acquired infection rate. We link the stochastic SEIS model to individual-level virological test results, capillary blood serology, as well as to the genomic data produced in this study. Modeled infection statuses were linked to virological test outcomes, and history of infection/vaccination was linked to capillary-blood serology accounting for time since infection/vaccination using the time-varying sensitivity estimates inferred with the statistical model. We use the model's explicit accounting of the origin in the force of infection that each individual experiences (within-class, between-class, community) to incorporate the genomic data in the model. Following methods used for transmission-pair reconstruction based on genetic distance information[58], we incorporate the pairwise phylogenetic distance between our study's sequences and contextual sequences from Geneva and Switzerland in the likelihood function using the relative strength of transmission rates from different sources the individuals experience, marginalizing over the unknown number of infection generations in the case of community infections. We draw inference using state-of-the art particle filtering methods that can account for coupling in dynamical systems[30,59].

**Scenario simulations.** The aim of the scenario simulations was to evaluate the potential impact of changes in the relative importance of within-school vs. community-acquired infections on the effectiveness of school-based interventions.

We use the dynamic model to produce simulations across scenarios of community-level transmission settings based on our fits to the SARS-CoV-2 Alpha, Delta, and Omicron periods, and across a range of reductions in within- and between-class transmission rates, from 0% (no change, i.e. baseline conditions) to 100% reduction (complete

interruption of transmission through that type of contact). We parametrize these scenarios in a general way which can represent different types of interventions that can either reduce all types of transmission rates such as masking or air ventilation, or specific contact types such as class cohorting where contacts between classes are restrained. We name "intervention scenario" the pair of within-class and between-class transmission reduction parameters.

For each transmission setting and intervention scenario, we performed 1000 simulations of 3 month-long transmission dynamics in a synthetic school population of 17 classes with 18 students per class and 3 adult staff. We base this number of classes in each school and the number of pupils per class on the sizes in our study population.

As interventions in schools may have a feedback effect on the reduction of community-wide transmission[21], we perform additional simulations considering the reductions in community-acquired infections for each transmission setting. Empirical estimates of the effect of school closure on community-wide transmission, the most stringent intervention scenario, during the COVID-19 pandemic ranged between no effect and 60%[20,60], with most estimates of effect ranging between 25% and 50%. For each intervention scenario, we consider a community-transmission scenario in addition to the baseline scenario (our inferred parameters), linking the reduction in the rate of community-acquired infections to the intensity of school-based interventions, where the reduction in community-wide transmission increases linearly from 0 (no interventions) to 50% reduction in community-acquired transmission rates at school closure. The reduction for each intervention scenario is computed assuming that within-class and between-class reductions have an equivalent effect on the reduction of community-wide transmission (see Supplementary Material Section S1.9 for details).

To assess intervention scenarios we compare simulation outcomes of the number of infections in the school population (whether from within-school or community transmission) to simulation outcomes from the baseline scenario (no interventions). For each intervention scenario, we further compare the simulation outcomes for the two scenarios of indirect reduction in community-wide interventions.

### Reporting summary
Further information on research design is available in the Nature Portfolio Reporting Summary linked to this article.

## Data availability
The epidemiologic data are available under restricted access for privacy protection reasons. Despite all the precautions taken, our data contain potentially identifying information due to the nature of this study and the fact that we are reporting information on relatively small groups of children from a specific geographic location. Therefore, the study steering committee members decided to make these data accessible to researchers who meet the criteria for access to confidential data upon reasonable request for data sharing to the Unit of Population Epidemiology (uep@hug.ch). All requests will be evaluated by the Data Access Committee and approved on the basis of their scientific quality. Requests will be evaluated within 3 months of submission, and access will be granted for a period of 1 year. The genetic sequences generated in this study have been deposited on GenBank (accession numbers given in Supplementary Table S9), and on GISAID at https://doi.org/10.55876/gis8.250626bw. A complete list of the labs that generated the sequence data used from GISAID can be found at https://doi.org/10.55876/gis8.240117ea and https://doi.org/10.55876/gis8.240117cm.

## Code availability
Code is available at https://github.com/UEP-HUG/serocov-schools-public.

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

## Acknowledgements

We gratefully acknowledge all data contributors, i.e., the authors and their originating laboratories responsible for obtaining the specimens, and their submitting laboratories for generating the genetic sequence and metadata and sharing via the GISAID Initiative, on which this research is based. We also gratefully acknowledge and thank all labs around the world that have collected and shared SARS-CoV-2 sequences we used in our study. We are very grateful to the children and teachers participating in the SEROCoV-Schools study and their families. We would like to thank the school staff for their responsiveness and support during the outbreak investigations. The authors also would like to thank the members of the Unit of Population Epidemiology at the Geneva University Hospitals for their daily support in all the tasks required by the SEROCoV-Schools study, and the Virology Laboratory staff of the Geneva University Hospital for performing the RT-PCRs and for helping with the storage of the samples. We thank Edward L. Ionides and Aaron A. King for support in the implementation of the particle filtering methods that enabled drawing inference from the mathematical model. The SEROCoV-Schools study was supported by the Swiss Federal Office of Public Health, the Private Foundation of the Geneva University Hospitals, the Fondation des Grangettes, the Center for Emerging Viral Diseases of the Geneva University Hospitals, and a Swiss National Science Foundation NRP (National Research Program) 78 Covid-19 Grant 198412 (to S.J.M., I.E.). The funders had no role in study design, data collection and analysis, decision to publish, or preparation of the manuscript.

## Author contributions

J.P.-S.: Conceptualization, Methodology, Software, Formal analysis, Writing - Original Draft, Visualization; M.B.: Investigation, Validation, Data Curation, Writing - Review & Editing; J.Le.: Writing - Original Draft, Supervision; JB: Investigation, Project administration, Writing - Review & Editing; E.H.: Software, Formal analysis, Validation, Writing - Review & Editing; G.M.: Investigation, Validation, Writing - Review & Editing; F.P.: Data Curation, Writing - Review & Editing; J.La.: Validation, Data Curation, Writing - Review & Editing; F.L.: Data Curation, Writing - Review & Editing; A.L.H.: Conceptualization, Writing - Review & Editing, Funding acquisition; K.P.-B.: Conceptualization, Writing - Review & Editing, Funding acquisition; S.M.: Supervision, Resources, Writing - Review & Editing, Funding acquisition; I.G.: Conceptualization, Writing - Review & Editing, Supervision, Funding acquisition; A.S.A.: Conceptualization, Methodology, Writing - Review & Editing, Supervision, Funding acquisition; I.E.: Conceptualization, Writing - Review & Editing, Supervision, Funding acquisition; S.S.: Conceptualization, Investigation, Resources, Writing - Review & Editing, Supervision, Funding acquisition; E.L.: Conceptualization, Investigation, Data Curation, Writing - Review & Editing, Supervision, Project administration.

## Competing interests

The authors declare no competing interests.

## Additional information

[1]Unit of Population Epidemiology, Department of Primary Care Medicine, Geneva University Hospitals, Geneva, Switzerland. [2]Center for Emerging Viral Diseases, Geneva University Hospitals and University of Geneva, Geneva, Switzerland. [3]Department of Epidemiology, Johns Hopkins Bloomberg School of Public Health, Baltimore, MD, USA. [4]Department of Medicine, Faculty of Medicine, University of Geneva, Geneva, Switzerland. [5]UNC Carolina Population Center, University of North Carolina at Chapel Hill, Chapel Hill, NC, USA. [6]Department of Epidemiology, University of North Carolina at Chapel Hill, Chapel Hill, NC, USA. [7]Institute of Social and Preventive Medicine, University of Bern, Bern, Switzerland. [8]Swiss Institute of Bioinformatics, Lausanne, Switzerland. [9]Multidisciplinary Center for Infectious Diseases, University of Bern, Bern, Switzerland. [10]Institute of Bioengineering, School of Engineering, Ecole Polytechnique Fédérale de Lausanne, Lausanne, Switzerland. [11]Laboratory of Virology, Diagnostics Department, Geneva University Hospitals, Geneva, Switzerland. [12]Department of Woman, Child and Adolescent Health, Pediatric Infectious Disease Unit, Geneva University Hospitals, Geneva, Switzerland. [13]Department of Pediatrics, Gynecology & Obstetrics, Faculty of Medicine, University of Geneva, Geneva, Switzerland. [14]Division of Primary Care, Geneva University Hospitals, Geneva, Switzerland. [15]Department of Health and Community Medicine, Faculty of Medicine, University of Geneva, Geneva, Switzerland. [16]Division of Tropical and Humanitarian Medicine, Geneva University Hospitals, Geneva, Switzerland. [17]School of Population and Public Health and Edwin S.H. Leong Centre for Healthy Aging, Faculty of Medicine, University f British Columbia, Vancouver, BC, Canada. [18]Université Paris Cité, Inserm, INRAE, Centre for Research in Epidemiology and Statistics Paris (CRESS), Paris, France. [19]Geneva School of Health Sciences, HES-SO University of Applied Sciences and Arts Western Switzerland, Geneva, Switzerland. [20]These authors contributed equally: Javier Perez-Saez, Mathilde Bellon. [21]These authors jointly supervised this work: Isabella Eckerle, Silvia Stringhini, Elsa Lorthe. ✉e-mail: javier.perez@hug.ch; elsa.lorthe@hug.ch

## the SEROCoV-Schools study group

Javier Perez-Saez [1,2,3,20] ✉, Mathilde Bellon[2,4,20], Julie Berthelot [1], Grégoire Michielin [10], Francesco Pennacchio [1], Julien Lamour[1], Florian Laubscher [11], Arnaud G. L'Huillier[11,12,13], Klara M. Posfay-Barbe[13], Sebastian J. Maerkl [10], Idris Guessous[14,15], Andrew S. Azman [2,3,16], Isabella Eckerle[2,4,21], Silvia Stringhini [1,15,17,21] & Elsa Lorthe [1,18,19,21] ✉

