## [Transparent Peer Review file · Nature Communications]

Evolving infectious disease dynamics shape school-based intervention effectiveness

Corresponding Author: Dr Javier Perez-Saez

Version 0:

Reviewer comments:

Reviewer #1

(Remarks to the Author)

Review for: Evolving infectious disease dynamics shape school-based intervention Effectiveness

Overview. This paper presents an analysis of SARS-CoV-2 rapid diagnostic test, RT-PCR tests and serological data collected during a reactive outbreak investigation in several schools in the Geneva area between March 2021 and February 2022. The data were analysed to establish the within-school-outbreak infection risk and the community-based infection risk using two separate methods: Firstly, a statistical model which used test positivity (RDT, PCR and serology) and information as to whether an outbreak had been reported in the class to infer the school-outbreak-associated and community-associated infection hazard rates. The second uses an individual based model of transmission, which incorporates genomic data by identifying 'likely infectors' and is fit to serology data using Block Particle Filtering (BPF). The key result of the paper is that although transmission is higher in schools than the community during the eras of earlier variants of SARS-CoV-2 this relative increase in transmission is reduced with successive waves of the pandemic. They conclude that this undermines the use of school-based NPIs, and that typical approach of introducing them when community prevalence is high is fundamentally flawed.

Overall comments on the manuscript:

This paper presents a very large body of work combining diverse data to provide additional evidence on an important question in the area of epidemic/pandemic control. I support the use of multiple models to analyse this data and evaluate its relevance to understanding transmission in schools vs the surrounding community.

The manuscript is very short and unfortunately, I found little in the text to support my understanding of the details of how the analysis was implemented, so most of my efforts were spent reading the supplement. The supplement offers more technical details, but lacks the structure and narrative required to give me a full understanding of how the different aspects of the models combine to adequately support the findings.

As the manuscript stands, I struggle to make a good review of the fidelity of the analysis and would like the opportunity to review a more complete manuscript if it is forthcoming. Until that time I am able to give only broad comments on the manuscript and the overall framing and findings of the study.

For this reason, I recommend including a much more thorough description of the models in the manuscript, which provides the reader with an intuition of what the purpose of the models is and at least a qualitative understanding of how they achieve this. For example:

- How are the two models combined as is suggested in the manuscript.
- The supplement presents an individual based model, but does not discuss how the population in the model is constructed, does this population directly reflect the school communities?
- The transmission model description includes a section on likely transmission pairs – but does not provide details of how it is integrated into the overall modelling framework.
- The manuscript discusses simulation studies, but the methods for these studies appear to be missing.

Overall comments on findings.

Under the assumption that the model fidelity is demonstrable, I have a few questions about the overall framing of this analysis and how it affects the key claims.

- Firstly, this data was generated as part of a contact tracing effort – which is, itself, a school based NPI, which would be expected to impact transmission. Have the authors considered the implications of this on their key observations about within school transmission? Do the models account for school-based exposure time vs. community-based exposure time given potential absence of children who test positive due to contact tracing efforts? It seems that these issues should at least be addressed in the manuscript.

- It does not seem surprising to me that if community prevalence is high then school-based transmission contributes less to the overall risk of infection of school aged children (this is likely also affected also by my first point, especially as community interventions and testing drastically reduced over the study period).

- It seems poorly conceived to me that school based NPIs are generally implemented when transmission is high in the community. In the case of COVID-19 most cases early in the pandemic school closures were implemented around the same time as stay-at-home orders, and schools were generally reopened before overall social distancing measures were lifted. If I am mistaken, I welcome evidence to the contrary.

- This analysis is based entirely on SARS-CoV-2 data, but the conclusions are broadly applied to school-based NPIs in general. This seems like a stretch, other pathogens (influenza for example) have very different dynamics in terms of age-specific infection rates. This should be thoroughly addressed discussion.

(Remarks on code availability)

The URL for the code did not work but I found it on the github. I have not reviewed thoroughly, the structure was quite tricky to parse immediately and there is no support in the readme for the user to understand how it is structured.

Reviewer #2

(Remarks to the Author)

With interest, I read the paper on the analysis of the COVID outbreak data among two primary schools and two preschools in Geneva from March 2021 to February 2022. It contains test data, serological data, outbreak data, and phylogenetic data. The authors combine these data to parameterize a transmission model, based on which they perform a scenario analysis regarding the implementation of NPIs.

I would like to request major revisions on the manuscript, perhaps mainly to clarify the methods or the way the authors speak about their undertaking and conclusions.

As the authors state, the main goal of school closures during an epidemic is to contribute to the control of transmission of a pathogen. Although not mentioned very clearly in the manuscript, I believe such control is achieved by reducing the effective reproduction number, ideally below 1. School closures contribute toward this goal because school closures force children to stay at home instead of sharing a class with 25 to 30 children, which thus leads to a severe reduction in the effective contacts between children. Subsequently, this reduction in (effective) contacts reduces the transmission of the pathogen. As long as school closure reduces the number of contacts between children, and children play a role in the transmission, there should be an effect on the reproductive number, and the transmission in the wider population should be (somewhat) reduced. If in a particular outbreak school closures could (let's say) reduce the reproductive number from 1.1 to 0.9, the impact on the outbreak would be large. If the impact is a reduction of the reproductive number from 5 to 4.8, the impact is less. But even in the latter, it still contributes towards the control of the pathogen. Nevertheless, school closures, and any other non-pharmaceutical intervention, are more effective when the numbers of cases and the resulting Force of Infection are low(er). Just because the risk of infection is directly linked to the number of cases, and thus you will have to work harder to reduce that risk when more cases are around. Furthermore, as school closures don't impact the susceptibility of children, school closures don't prevent cases per se, but rather slow down transmission. The exact impact of school closures (or other sorts of NPIs) on the reproductive number would depend on many aspects within the outbreak.

The authors seem to conclude something similar to the above. However, I was somewhat confused by their approach, the generalizability, and the language they use. Their main conclusion seems to be that "While there may be valid reasons for maintaining school-based interventions during periods of high epidemic activity, such as protecting vulnerable students or addressing staffing shortages, epidemic control is not one of them." And they call for studies to elucidate the "NPI-tipping point." I somewhat understand what they mean, but I found their argument hard to distill from the text. What exactly do the authors mean by control? Do they mean that school closures cannot contribute to the control? Thus, that had no effect whatsoever on the effective R? Or that given the specifics of SARS-CoV-2 transmission in Swiss society at that time, school closures alone did not push the R below 1? Or that the risk for infection was already so high that there was a limited effect, especially because of the specifics of SARS-CoV-2 transmission among children and adults? But to what extent would this also be true for mask-wearing and any other NPI? Hopefully, the authors understand my confusion and could improve the clarity and reasoning of their methods, results, and discussion to make this more clear. Furthermore, I was a bit confused about the premise and the chosen endpoints in the scenario analysis. Do the authors look at the infections in the whole population, and thus community transmission includes transmission from/to other age groups, or only in school-aged children? And do they aim to reduce the school-originated infections only? Or aim to reduce all infections? Furthermore, the model scenario is run for 3 months, with on the y-axis "averted infections," where these cases are effectively not averted, but delayed. Furthermore, in general, not a single NPI is believed to be able to control transmission by itself, and therefore

multiple NPIs are used together—and it is the challenge to find a combination between the number and intensity on one hand, and still reducing the transmission sufficiently to gain control of the outbreak on the other. I missed this aspect in their discussion, which I believe is very important when speaking of “tipping points” of individually defined NPIs (with all the problems of the exact definition and the effectiveness of such NPI in practice).

(Remarks on code availability)

Reviewer #3

(Remarks to the Author)

The authors report on a school based study over 2 years of SARS-CoV-2 circulation. They use a mix of clinical, biological and genetic data to estimate parameters related to transmission in schools and illustrate that over successive waves, the part of in-school transmission reduced relative to introductions from elsewhere. Their main conclusion is that school based interventions may have less impact when outside circulation is high and call for an assessment depending on the context.

Overall, the integration of many sources is commendable. It provides a detailed understanding on how transmission in schools was shaped during these COVID-19 waves.

Regarding the manuscript, I'm always uncomfortable with papers based on heavy modelling where all model details are in supplementary information. I believe an overall description (showing the breakdown of the likelihood, for example) adds to the understanding of the methods.

Maybe this could be explained in a little more detail.

Regarding the investigation, it's not entirely clear why there is a "statistical" model then a "dynamical" model. It seems that both use the same data and model essentially the same thing, with more data in the dynamical model and a better model for transmission during outbreaks. The justification given in S1.8 is rather poor. Why can't you estimate θ_{sero} from the second model ? As the model for outbreaks is not the same in the 2 approaches, how do you decide that the estimate from the first model is compatible with the second ?

While I don't doubt that sustained external introductions from the community makes school-based interventions less impactful in schools, nor that context dependent assessment is a good idea, I'm not sure that the authors provide much evidence that school-based interventions could not provide control when "community" transmission is high. Indeed, the conclusion of the paper is questionable due to the definition and simulation of "community" incidence to represent school based interventions.

Here, the authors perform simulations with community incidence unaffected by school-based interventions to illustrate their conclusion in Fig 4C.

However, one may wonder how much sustained activity would remain in the community if school-based interventions were generalized.

In the data presented, "community" incidence reflects, among others, epidemic activity in the schools not included in this study. As incidence in schools is much higher than in the community, it would presumably change if school-based interventions were generalized, in a proportion linked to the contribution of children to transmission - disease and wave dependent.

Consequently, the simulations basing the "counterintuitive" main result may correspond to the very situation that would be avoided with the adoption of school based interventions.

Only a whole population model could provide more insight into that (and this has been done a lot during the COVID-19 pandemic).

minor points :

M1 - "Further, SARS-CoV-V-2 RT-PCR testing was encouraged..." should it be SARS-CoV-2 ?

M2 - On Figure 1, the empty dots are barely visible, and their colour hard to see. Full dots would certainly be better.

M3 - SI - S9. (ref NBA, Alpha paper in England) please provide correct reference

M4 - S1.8.1 how were dates of epidemic defined ? Also a table with the t_L , t_R could be provided

M5 - S1.8.1.2.1,21,3: There is some confusion between implementation and model here. As a result, it makes it difficult to see whether the model is correct. The authors should try to present the probabilistic model rather than STAN tricks they used to implement it.

The model could be something like $P(Y_{it}, Z_{it_test}, t_{iinf})$ where Y_{it} is the serology result and Z_{it_test} the result of the RADT. that you split as $P(Y_{it} | t_{iinf}=t_{test}, Z_{it_test}) P(Z_{it_test} | t_{iinf}) P(t_{iinf})$

.1 $(P(Y_{it} | Z_{it}=1, T_{inf}))$ First, what is this "multinomial" for y^* ? in a multinomial, the sum of probabilities should be 1, it is not the case for p_1 and p_2 .

You mention t_{test} and t_{inf} , but we don't know what is the relation between them : is the first taken to be a proxy of the second ?

What happens if there are several t_{test} for the same person, and then how would it affect the computation?

When you mean earliest here, is it the earliest in the follow-up or the last one before the serology ?

.2: same as before, what is the notation Bernoulli() ? Shouldn't this be $P(y=1) = \phi(t) \sum_j (\theta^{t-t_j} P(t_j)) + (1-\phi(t))$ $(1-\theta^{t-t_j})$

.3: This is $P(z_{it} | t_{inf})$

M6 - S2.5 - the model does not seem to match serology at baseline. Why is it so and what are the consequences of that?

(Remarks on code availability)

Version 1:

Reviewer comments:

Reviewer #1

(Remarks to the Author)

Thank you for your thorough responses to my comments from the initial round of review. I am happy to see the methods generously expanded and feel that the paper is now much easier to understand. I have some remaining points that I would like to raise.

1. I still struggle a little with the framing of the conclusions. And from your responses I think you speak more clearly about the results. What you have measured is a reduction in the risk contribution of school-based infectious exposures as total prevalence increases. It seems to me that the discussion of different VOCs is truly just a proxy for periods of varying community prevalence. It might therefore be clearer to discuss your results as the former rather than starting with 'successive VOCs' as the chief explanatory component.

2. I also want to come back to the point about generality. I disagree with the point about total prevalence always reducing the impact of school closures. In the instance where transmission is much higher amongst school children this would not necessarily be the case, or at least the relevance of this would be substantially reduced, for example in the context of an influenza pandemic where age-dependent immunity profiles play an important part in transmission dynamics (e.g. 2009). I recommend that this should be addressed directly in the discussion.

3. Finally, reading the paper from the start, I still find it hard to glean precisely how each result was acquired, i.e. without skipping to the methods. I think a sentence to put each result in context (methodologically) would support understanding here.

(Remarks on code availability)

Reviewer #2

(Remarks to the Author)

I am not entirely satisfied with how my comments have been incorporated into the revised version. My core argument was that school-based interventions should be understood as part of a broader package of measures aimed at reducing transmission across the entire population. The ultimate goal of such interventions is population-level protection. From the outset, the rationale behind these non-pharmaceutical interventions (NPIs) has been their contribution to lowering the effective reproduction number (R). Specifically, for school-based interventions, this has always been about assessing how schools contribute to the effective R. If their contribution is small, the interventions within schools will only have a limited impact, as transmission is largely driven by factors outside of schools. This point has been clear from the beginning. Hence, I feel the authors still need to better articulate how their modelling and results contribute new insights within this established framework. As it stands, the text still creates confusion for me. It still appears to me that the authors first isolate school-based interventions from their overarching objective and focus narrowly on their ability to mitigate the burden within schools. However, this reasoning is for me still missing the point. As a result, I would like to request that the authors revise their discussion to a greater extent and reflect more explicitly what new insights their work provides. Specifically, it could highlight the lessons learned about estimating the contribution of schools to the overall effective R. For example, how might we define when the transmission originating from schools is "too low" warranting an exclusion of school based NPIs to protect the overall population? Additionally, the authors could propose frameworks, methodological approaches etc.

(Remarks on code availability)

Reviewer #3

(Remarks to the Author)

I thank the authors for the added work.
I have no further comment.

(Remarks on code availability)

Version 2:

Reviewer comments:

Reviewer #1

(Remarks to the Author)

First, thanks for your thorough responses. Overall, I am happy with the changes and it is now much easier to follow.. My only remaining request is that you tone down the sentence in the discussion:

"These simulations demonstrate that school-based interventions can be highly effective when community transmission is low and thus community-acquired infections by school-aged children rare, but can be largely ineffective when community transmission rates are high."

I think a more appropriate phrasing might be along the lines of:

"These simulations demonstrate that school-based interventions can be highly effective when community transmission is low and thus community-acquired infections by school-aged children rare. However, when community transmission rates are high, school-based interventions can only be implemented effectively as part of a comprehensive suite of interventions that target transmission in multiple settings."

(Remarks on code availability)

Reviewer #2

(Remarks to the Author)

I thank the authors for their clarifications. But I am sorry, I respectfully note that I believe I dispute their premise.

They write: "The premise of our work is that to better address this issue, it is key to understand within-school transmission dynamics to assess the eventual effect of NPIs in schools. In fact, any modelling study assessing the role of school-based NPIs on epidemic progression will need to make assumptions on how they will change infection patterns in school-aged populations and their contribution to community-level transmission through contact networks."

While I appreciate the authors' perspective, I do not believe this is necessarily the case. Network-based transmission models can analyze the propagation of disease within a population, even without explicitly modelling the within-school dynamics. Such models can nonetheless draw conclusions about the potential contribution of schools to overall transmission and, by extension, about the possible impact of school-based NPIs.

For example, population-level transmission patterns would be markedly different depending on whether schools contribute significantly to transmission or not. The timing and spatial distribution of infections among adults, for instance, would differ between scenarios where schools act as a major transmission node versus scenarios where they do not. This distinction arises because transmission pathways originating in schools differ from those originating in other settings, such as workplaces.

However, I do agree with "their contribution to community-level transmission through contact networks". You need to understand transmission patterns within families to make a quick assessment regarding the contribution of schools. When children often infect their parents, school based NPIs are a good idea, when it is the other way around – and parent bring infections to households – this is less likely to be the case.

But when I dispute their premise I don't believe I dispute their work – only the discussion. Hence, I don't think my point would justify to block this publication further. I just hope that the authors could be more explicit about the need to explicitly model within school dynamics to address this particular question of the impact of NPIs within a pandemic.

(Remarks on code availability)

NA

REVIEWER COMMENTS

Reviewer #1 (Remarks to the Author):

[1.1] Overview. This paper presents an analysis of SARS-CoV-2 rapid diagnostic test, RT-PCR tests and serological data collected during a reactive outbreak investigation in several schools in the Geneva area between March 2021 and February 2022. The data were analysed to establish the within-school-outbreak infection risk and the community-based infection risk using two separate methods: Firstly, a statistical model which used test positivity (RDT, PCR and serology) and information as to whether an outbreak had been reported in the class to infer the school-outbreak-associated and community-associated infection hazard rates. The second uses an individual based model of transmission, which incorporates genomic data by identifying 'likely infectors' and is fit to serology data using Block Particle Filtering (BPF). The key result of the paper is that although transmission is higher in schools than the community during the eras of earlier variants of SARS-CoV-2 this relative increase in transmission is reduced with successive waves of the pandemic. They conclude that this undermines the use of school-based NPIs, and that typical approach of introducing them when community prevalence is high is fundamentally flawed.

Overall comments on the manuscript:

This paper presents a very large body of work combining diverse data to provide additional evidence on an important question in the area of epidemic/pandemic control. I support the use of multiple models to analyse this data and evaluate it's relevance to understanding transmission in schools vs the surrounding community.

We thank this reviewer for the positive appreciation of our work.

[1.2] The manuscript is very short and unfortunately, I found little in the text to support my understanding of the details of how the analysis was implemented, so most of my efforts were spent reading the supplement. The supplement offers more technical details, but lacks the structure and narrative required to give me a full understanding of how the different aspects of the models combine to adequately support the findings.

As the manuscript stands, I struggle to make a good review of the fidelity of the analysis and would like the opportunity to review a more complete manuscript if it is forthcoming. Until that time I am able to give only broad comments on the manuscript and the overall framing and findings of the study.

We acknowledged that the original version of the main text was dense, and have significantly expanded it in response to all three Reviewer's requests as detailed below. We hope that this will help in the evaluation of our results and conclusions.

For this reason, I recommend including a much more thorough description of the models in the manuscript, which provides the reader with an intuition of what the purpose of the models is and at least a qualitative understanding of how they achieve this. For example:

[1.3] - How are the two models combined as is suggested in the manuscript.

In this study we pursue two aims using modeling, the first being inferential modeling and the second strategic/scenario modeling. The statistical and mathematical models address the inferential

goal, and the scenario modeling goal leverages the fit dynamic model. The two models were fit independently to the data, and the only direct link between them is that we infer the time-varying sensitivity of the capillary-blood serology assay using the statistical model which we feed as an input to compute the likelihood function in the dynamic model.

We have expanded the Methods section on the modeling analyses to better clarify these two goals and the ways the models were combined:

“This study had two main goals: i) to infer the the relative contributions of within-school transmission vs. community importation of SARS-CoV-2 in the study population and how these changed in time (inferential modeling goal), ii) and building on these inferential results, to investigate the implications of changing strength of community importations on the effectiveness of school-based interventions against respiratory pathogens (strategic/scenario modeling goal). We addressed our study goals by developing two modeling frameworks: a statistical model and a dynamic transmission model. Full model details are given in the Supplementary Material Section S1.8.

We leveraged both the statistical and dynamic modeling frameworks for the inferential modeling goal because of their complementarity in the targets of inference they were equipped to pursue, and to strengthen the robustness of our results. The common target of inference of both modeling frameworks were i) the cumulate attack rate during SARS-CoV-2 outbreaks in our study schools which could not be evaluated directly because of data missingness, and ii) the community-imported transmission rate and how it changed with SARS-CoV-2 VOCs. A key target of inference of the statistical model was the time-varying sensitivity of the capillary-blood serological assay we used in this study, which had not been previously evaluated with time post-infection. These estimates of time-varying sensitivity were in turn an input to evaluate the likelihood function of the dynamic model. By explicitly representing SARS-CoV-2 transmission events, the dynamic model allowed to refine inference on within-school vs. community by incorporating the genomic data produced in this study, and further differentiating between within-class and between-class transmission.

The dynamic modeling framework allowed to pursue the strategic/scenario modeling goal on the effectiveness of school-based interventions by explicitly manipulating the within-class and between-class transmission rates. Using inferred parameters as baselines, we investigated the effect of reductions in within-school transmission rates on overall attack rates under different scenarios of community-acquired infections.”

[1.4] - The supplement presents an individual based model, but does not discuss how the population in the model is constructed, does this population directly reflect the school communities?

Within the SeroCoV-Schools study we recorded information on the class composition (pupils and staff) of all classes in the enrolled study schools, we were therefore able to simulate the whole study school population in the dynamic mode whether individuals were enrolled or not. We here show the distribution of class sizes by study school to illustrate the data.

Figure A: Histogram of class size by educational setting in the SeroCoV-Schools study.

[1.5] - The transmission model description includes a section on likely transmission pairs – but does not provide details of how it is integrated into the overall modelling framework.

In the dynamic model we explicitly model the force of infection to which each individual is subject to as a function of the infectious status of other class and school members, in addition to community-acquired infections. As such, the model accounts for all possible transmission pairs from infectious to susceptible individuals, accounting for the contact network in each school that distinguishes between within-class and between-class contacts (Section S1.8.2.1 of the Supplementary Material).

The likeness of transmission pairs is accounted for in the genetic component of the likelihood function (Section S1.8.2.7.3 of the Supplementary Material). This genetic likelihood component is evaluated only for those participants for which a valid SARS-CoV-2 genetic sequence was available. The likeliness of transmission pairs directly appears in the likelihood function through the probability that each infector with available genomic data was at the origin of the infection, which is a function of the transmission rate parameters of each type of contact (within-class vs. between class vs. community-acquired). The likelihood of transmission pairs given their phylogenetic distance therefore directly informs the relative magnitude of the transmission rate parameters.

We have made this clearer in the new Methods section describing inference using the dynamical model:

“We use the model’s explicit accounting of the origin in the force of infection that each individual experiences (with-class, between-class, community) to incorporate the genomic data in the model. Following methods used for transmission-pair reconstruction based on genetic distance information, we incorporate the pairwise phylogenetic distance between our study’s sequences and contextual sequences from Geneva and Switzerland in the likelihood function using the relative strength of transmission rates from different sources the individuals experience, marginalizing over the unknown number of infection generations in the case of community infections.”

[1.6] - The manuscript discusses simulation studies, but the methods for these studies appear to be missing.

We have added a section in the Methods on the description of the simulation studies:

“The aim of the scenario simulations was to evaluate the potential impact of changes in the relative importance of within-school vs. community-acquired infections on the effectiveness of school-based interventions.

We use the dynamic model to produce simulations across scenarios of community-level transmission settings based on our fits to the SARS-CoV-2 Alpha, Delta, and Omicron periods, and across a range of reductions in within- and between-class transmission rates, from 0% (no change, i.e. baseline conditions) to 100% reduction (complete interruption of transmission through that type of contact). We parametrize these scenarios in a general way which can represent different types of interventions that can either reduce all types of transmission rates such as masking or air ventilation, or specific contact types such as class cohorting where contacts between classes are restrained. We name “intervention scenario” the pair of within-class and between-class transmission reduction parameters.

For each transmission setting and intervention scenario, we performed 1,000 simulations of 3 month-long transmission dynamics in a synthetic school population of 17 classes with 18 students per class and 3 adult staff. We base the number of classes in each school and the number of pupils per class on the sizes in our study population.

As interventions in schools may have a feedback effect on the reduction of community-wide transmission, we perform additional simulations considering the reductions in community-acquired infections for each transmission setting. Empirical estimates suggest that school closure, the most stringent intervention scenario, during the COVID-19 pandemic reduced community-wide transmission ranged between no effect and 60%, with most estimates of effect ranging between 25% and 50%. For each intervention scenario, we consider a community-transmission scenario in addition to the baseline scenario (our inferred parameters), linking the reduction in the rate of community-acquired infections to the intensity of school-based interventions: one where the reduction in community-wide transmission increases linearly from 0 (no interventions) to 50% reduction in community-acquired transmission rates at school closure. The reduction for each intervention scenario is computed assuming that within-class and between-class reductions have an equivalent effect on the reduction of community-wide transmission (see Supplementary Material Section S1.9 for details).

To assess intervention scenarios we compare simulation outcomes of the number of infections in the school population (whether from within-school or community transmission) to simulation outcomes from the baseline scenario (no interventions). For each intervention scenario, we further compare the simulation outcomes for the two scenarios of indirect reduction in community-wide interventions.”

We have also added technical details in Supplementary Material Section S1.9.

Overall comments on findings.

Under the assumption that the model fidelity is demonstrable, I have a few questions about the overall framing of this analysis and how it affects the key claims.

[1.7] - Firstly, this data was generated as part of a contact tracing effort – which is, itself, a school based NPI, which would be expected to impact transmission. Have the authors considered the implications of this on their key observations about within school transmission? Do the models account for school-based exposure time vs. community-based exposure time given potential absence of children who test positive due to contact tracing efforts? It seems that these issues should at least be addressed in the manuscript.

We thank this Reviewer for this interesting point. We agree that the reactive testing strategy in our study may have impacted within-school transmission as quarantine was compulsory during the first part of the study period (March 2021-Feb 2022). In fact our case identification led to class closure during one outbreak in April-May 2021 (two classes, totalling 40 children, closed for 10 days starting April 30, 2021). However, as we had information on SARS-CoV-2 test results (both from our study and from the central test repository of the state of Geneva) and school measures we were able to account for student and staff absences due to quarantine and directly in our modeling frameworks. We therefore believe that our parameter inferences were not affected by the direct effect of our testing protocol, in particular the relative importance of within-school vs. community-acquired infections.

It may however be possible that some school-wide measures might have been taken in our study schools that would not have occurred if our study had not taken place, or that adherence to measures may have been lower in the absence of our study. We acknowledge that this indirect effect may have had an effect on our inferences which we cannot control for, but we however do not believe that this effect would have been large enough to undermine the main conclusions of our work regarding the relative importance of within-school vs. community-acquired infections.

We have added a sentence in the limitations section of the discussion to mention this point:

“Finally, our inferential estimates on within-school transmission might be influenced by the indirect effect of our study on school control measure implementation and adherence, which we cannot control for. It is however unlikely that this eventual indirect effect is large enough to undermine our main conclusions on the relative importance of within-school vs. community-acquired transmission.”

[1.8] - It does not seem surprising to me that if community prevalence is high then school-based transmission contributes less to the overall risk of infection of school aged children (this is likely also affected also by my first point, especially as community interventions and testing drastically reduced over the study period).

We agree with the intuitiveness of this result. We however would like to stress two points related to the novelty of this work in terms of quantifying how big of an impact community-acquired transmission has on the effectiveness of school-based interventions.

First, to our knowledge this is the first quantification of the magnitude within-school vs. community-acquired SARS-CoV-2 transmission in schools, and its changes with successive VOCs during the COVID-19 pandemic.

Second, these novel inferential results allow in turn to set informed parameter ranges in our strategic/scenario modeling simulations to investigate the effectiveness of school-based interventions. For instance, previous scenario modeling studies on the effect of school-based NPIs only considered observed daily community infection rates of up to 50 cases per 100,000, while in the Omicron period of our study infections rose to more than 2,000 per 100,000 per week (Giardina et al. 2022).

We therefore believe that the value of this study lies in quantifying the intuitive notion of reduced school-based interventions in different settings of community transmission by anchoring scenario simulations in robust parameter estimates based on a rich dataset.

[1.9] - It seems poorly conceived to me that school based NPIs are generally implemented when

transmission is high in the community. In the case of COVID-19 most cases early in the pandemic school closures were implemented around the same time as stay-at-home orders, and schools were generally reopened before overall social distancing measures were lifted. If I am mistaken, I welcome evidence to the contrary.

We thank this Reviewer for this comment. We here provide evidence supporting our statement that school closure decisions during the COVID-19 pandemic did in fact occur during periods of high transmission, especially during the VOC period (2021-2022).

To assess the conditions under which school closures occurred we collated a global dataset of publicly-available national data on school closure dates, reported incidence and changes in the effective reproduction number (Figure A below). Details are provided in Supplementary Material Section S1.10. We find that in 2020, school closures did indeed occur in periods of low incidence, as suggested by this Reviewer, although there was substantial heterogeneity by continent. However, in 2021 and especially in (Omicron-dominated) 2022 school closures occurred much more frequently in periods of high and increasing incidence (Figure B).

We have added these data to the Supplementary Material, and referenced it in the Discussion. We have also nuanced the sentence on the frequency of school-based NPIs during periods of high transmission as:

*“This contrasts with how school-based NPIs have **tended to have been implemented, especially during the VOC-dominated phase of the COVID-19 pandemic (Supplementary Figure S15), where high community transmission typically triggers mitigation measures up to including school closures.**”*

Figure B: Analysis of school closure timings with respect to COVID-19 case incidence and effective reproduction number. a) Each point represents a school closure, and its position indicates the value of the effective reproduction number on the date of school closure (x axis), and the relative level of community incidence in that year (y axis). The relative incidence level is shown in terms of the year-standardized square root of the average weekly incidence on the

date of school closures, where the value of 1 indicates 1 standard deviation away from the yearly mean weekly incidence. The upper-right quadrant defined by the dotted lines defines school closures that occurred in COVID-19 incidence periods: more than 1sd from the mean average weekly incidence and $R_{eff} > 1$. b) Proportion of school closures that occurred by community incidence level.

[1.10] - This analysis is based entirely on SARS-CoV-2 data, but the conclusions are broadly applied to school-based NPIs in general. This seems like a stretch, other pathogens (influenza for example) have very different dynamics in terms of age-specific infection rates. This should be thoroughly addressed discussion.

We agree with this Reviewer that other respiratory pathogens have different natural histories and transmission dynamics, as well as age-specific infection rates. Although we concede that these differences may affect the quantitative outcomes of our scenario simulation comparison, we believe that the general result of our study is robust to these variations, ie. that if community-wide transmission of a respiratory pathogen is too high it will undermine the effectiveness of school-based interventions. As noted in Reviewer comments [1.8] and [2.3], this statement may seem intuitive, but we believe it has important public health value given the empirical evidence we present above on the timing of school closures during the COVID-19 pandemic that stand in contradiction with this result.

We have updated the Discussion to clarify these points:

“Finally, the results and conclusions of our scenario simulation investigations on the use of school-based measures in varying transmission settings are anchored in our study of SARS-CoV-2, and the same type of comparison may yield different quantitative results using natural history parameters and age-specific transmission rates for other respiratory pathogens. We however expect that our main result, ie. that school-based interventions lose effectiveness when community transmission is too high, is robust to this variance and applies across infectious diseases involving school-aged children. The clear definition of what “too high” remains subject of future work for specific pathogens and transmission settings of public health importance.”

A point indirectly linked to this comment is that whether school-based interventions are considered in the first place may be specific to each pathogen, for instance if schools are not believed to play an important role in community-wide epidemic dynamics and are thus not relevant targets of interventions. The results of our study are in turn possibly less relevant to this type of public health context.

Reviewer #1 (Remarks on code availability):

The URL for the code did not work but I found it on the github. I have not reviewed thoroughly, the structure was quite tricky to parse immediately and there is no support in the readme for the user to understand how it is structured.

We apologize for the broken url which is now fixed. We have updated the Readme to clarify the aims and content of each script in the repo.

Reviewer #2 (Remarks to the Author):

[2.1] With interest, I read the paper on the analysis of the COVID outbreak data among two primary schools and two preschools in Geneva from March 2021 to February 2022. It contains test data, serological data, outbreak data, and phylogenetic data. The authors combine these data to parameterize a transmission model, based on which they perform a scenario analysis regarding the implementation of NPIs.

We thank this reviewer for the in-depth appreciation of our work.

[2.2] I would like to request major revisions on the manuscript, perhaps mainly to clarify the methods or the way the authors speak about their undertaking and conclusions.

We have addressed the points raised by the reviewer below, which we believe have improved the clarity of the manuscript.

[2.3] As the authors state, the main goal of school closures during an epidemic is to contribute to the control of transmission of a pathogen. Although not mentioned very clearly in the manuscript, I believe such control is achieved by reducing the effective reproduction number, ideally below 1. School closures contribute toward this goal because school closures force children to stay at home instead of sharing a class with 25 to 30 children, which thus leads to a severe reduction in the effective contacts between children. Subsequently, this reduction in (effective) contacts reduces the transmission of the pathogen. As long as school closure reduces the number of contacts between children, and children play a role in the transmission, there should be an effect on the reproductive number, and the transmission in the wider population should be (somewhat) reduced. If in a particular outbreak school closures could (let's say) reduce the reproductive number from 1.1 to 0.9, the impact on the outbreak would be large. If the impact is a reduction of the reproductive number from 5 to 4.8, the impact is less. But even in the latter, it still contributes towards the control of the pathogen. Nevertheless, school closures, and any other non-pharmaceutical intervention, are more effective when the numbers of cases and the resulting Force of Infection are low(er). Just because the risk of infection is directly linked to the number of cases, and thus you will have to work harder to reduce that risk when more cases are around. Furthermore, as school closures don't impact the susceptibility of children, school closures don't prevent cases per se, but rather slow down transmission. The exact impact of school closures (or other sorts of NPIs) on the reproductive number would depend on many aspects within the outbreak.

We thank this Reviewer for a clear presentation of their interpretation of our work.

[2.4] The authors seem to conclude something similar to the above. However, I was somewhat confused by their approach, the generalizability, and the language they use. Their main conclusion seems to be that "While there may be valid reasons for maintaining school-based interventions during periods of high epidemic activity, such as protecting vulnerable students or addressing staffing shortages, epidemic control is not one of them." And they call for studies to elucidate the "NPI-tipping point." I somewhat understand what they mean, but I found their argument hard to distill from the text. What exactly do the authors mean by control? Do they mean that school closures cannot contribute to the control? Thus, that had no effect whatsoever on the effective R? Or that given the specifics of SARS-CoV-2 transmission in Swiss society at that time, school closures alone did not push the R below 1? Or that the risk for infection was already so high that there was a limited effect, especially because of the specifics of SARS-CoV-2 transmission among children and adults?

We thank this reviewer for raising these points, which we clarify below.

By control we mean the reduction of infection attack rates in the school-based population and in their surrounding community below specific public health targets, which may be bringing the effective reproduction number (R_{eff}) below 1 or reaching specific incidence targets in given age groups. In this sense we do not intend that school-based interventions will have no effect on school-aged infection rates and community-wide transmission (0 effect on R_{eff}), but rather that the effect will probably be too small to reach the intended public health target for epidemic control. We further did not intend to infer the specific effect of school-based interventions, including school closures, on the COVID-19 pandemic in Switzerland, which others have provided estimates for (Banholyer et al., 2021), and these have been shown to be context-dependent (Banholzer et al., 2022).

The last sentence of this comment better reflects our intention regarding SARS-CoV-2 and the study period. The aim of this sentence was however more to give a clear forward-looking message on the implications of the work, rather than an assertion of how things unfolded in Switzerland and how they could have been with a counter-factual perspective. We have clarified this in the Discussion:

“While there may be valid reasons for maintaining school-based interventions during periods of high epidemic activity, such as protecting vulnerable students or addressing staffing shortages, epidemic control in terms of reductions of infections in school-aged populations may not be within practical reach.”

[2.5] But to what extent would this also be true for mask-wearing and any other NPI? Hopefully, the authors understand my confusion and could improve the clarity and reasoning of their methods, results, and discussion to make this more clear.

The focus of our work was on the implementation of NPIs in school settings, with school closure being an extreme. In this sense masking policy in schools falls within the types of interventions covered by our general implementation of the scenario modeling analysis (see response to point [1.6]). We however expect that similar conclusions would hold for other types of NPIs that either target specific settings in isolation (for instance measures targeting nursing homes), or secondary transmission events (eg. contact tracing).

We have added a sentence in the Discussion to highlight this point:

“More generally, our conclusions on the sensitivity of NPI effectiveness as a function of community-wide transmission intensity may extend to other types of epidemic control interventions such as contact tracing, thus highlighting the importance of relevant population-level epidemic indicators.”

[2.6] Furthermore, I was a bit confused about the premise and the chosen endpoints in the scenario analysis. Do the authors look at the infections in the whole population, and thus community transmission includes transmission from/to other age groups, or only in school-aged children?

We provide a more thorough description of the scenario modeling methods in our updated manuscript (see response to comment [1.6]). As the focus of our study was on within-school pathogen transmission and control, our endpoints focus on the school-aged population in the modeled synthetic schools, and not in infections in the whole population.

[2.7] And do they aim to reduce the school-originated infections only? Or aim to reduce all infections?

In our original scenario simulation results we only focused on the effects of NPIs on reducing school-originated infections, as we did not consider feedback effects between school transmission dynamics and community-wide transmission.

In response to Reviewer 3 we have now added additional analyses exploring possible feedback effects between school-based interventions and community-wide transmission, as detailed below.

[2.8] Furthermore, the model scenario is run for 3 months, with on the y-axis “averted infections,” where these cases are effectively not averted, but delayed.

We agree with this reviewer that the extent to which infections are effectively averted during the course of an outbreak or delayed effectively depends on the reduction of the transmission rates (when R_{eff} remains above one), and that some cases in the intervention scenarios may occur after the 3 month period. We chose 3 months to mirror the typically short length of period of relative stability of intervention measures and VOCs during the COVID-19 pandemic, and thus of transmission rates, which also seemed reasonable time horizons for public health decision making.

To answer this concern, we have added a sensitivity analysis running the scenarios simulations for 6 and 12 months instead of the 3 months used in our original analysis. As illustrated in Figure C below (now supplementary Figure S14), the duration of simulations does sensibly reduce the computed effectiveness of interventions. However this effect applies similarly across community transmission scenarios, and therefore does not affect our main result, ie. that in the setting of high community transmission school-based may have low effectiveness.

We have now added this sensitivity analysis to the Methods section and to the Supplementary Material in section S1.9.2 and Figures S12-S14.

Figure C: Sensitivity analysis of scenario simulations along the diagonal of scenario space by simulation duration in

days. Legend as in Figure 4b of the main text.

We now mention the robustness of our results in the Results section:

“These results were robust to different durations of epidemic simulations (6 and 12 months) (Supplementary Figure S12-S13), and when accounting for feedback between reductions in within-school and community-wide transmission (Supplementary Figures S12, S14)”

[2.9] Furthermore, in general, not a single NPI is believed to be able to control transmission by itself, and therefore multiple NPIs are used together—and it is the challenge to find a combination between the number and intensity on one hand, and still reducing the transmission sufficiently to gain control of the outbreak on the other. I missed this aspect in their discussion, which I believe is very important when speaking of “tipping points” of individually defined NPIs (with all the problems of the exact definition and the effectiveness of such NPI in practice).

We thank this Reviewer for this point, and agree that a single NPI is unlikely to be sufficient to reach public health aims in isolation, as seen during the COVID-19 pandemic. As mentioned above, we however believe that our results hold value for the consideration of other NPIs than school-based interventions, and that the concept of NPI “tipping point” retains its potential relevance even when considering packages of NPIs. Because the main focus of our study was on school-based interventions, which are particularly controversial, we had decided to narrow the discussion on the consideration of these.

We have now added a sentence to mention the consideration of NPI packages beyond single school-based interventions:

“Beyond school-based interventions, NPIs are typically considered in packages which may further modify the relation between epidemiologic contexts and the effectiveness of single NPIs, thus highlighting the importance of relevant population-level epidemic indicators to inform public health decision making.”

Reviewer #3 (Remarks to the Author):

[3.1] The authors report on a school based study over 2 years of SARS-CoV-2 circulation. They use a mix of clinical, biological and genetic data to estimate parameters related to transmission in schools and illustrate that over successive waves, the part of in-school transmission reduced relative to introductions from elsewhere. Their main conclusion is that school based interventions may have less impact when outside circulation is high and call for an assessment depending on the context.

Overall, the integration of many sources is commendable. It provides a detailed understanding on how transmission in schools was shaped during these COVID-19 waves.

We thank this reviewer for the careful consideration of our work.

[3.2] Regarding the manuscript, I'm always uncomfortable with papers based on heavy modelling where all model details are in supplementary information. I believe an overall description (showing the breakdown of the likelihood, for example) adds to the understanding of the methods. Maybe this could be explained in a little more detail.

As detailed in response to Reviewer comments [1.3]-[1.5]-[1.6], we have significantly expanded the modeling section of the Methods section of the Manuscript, to which we refer this Reviewer to. We believe that these changes have significantly improved the clarity of the manuscript.

[3.3] Regarding the investigation, it's not entirely clear why there is a "statistical" model then a "dynamical" model. It seems that both use the same data and model essentially the same thing, with more data in the dynamical model and a better model for transmission during outbreaks. The justification given in S1.8 is rather poor. Why can't you estimate θ_{sero} from the second model? As the model for outbreaks is not the same in the 2 approaches, how do you decide that the estimate from the first model is compatible with the second?

We refer the Reviewer to the response to comment [1.3] above for a clearer explanation of the rationale, uses and integration of the two modeling frameworks.

Regarding the estimation of the sensitivity of the capillary-blood serological assay, we had considered estimating it jointly in the dynamic modeling framework which seemed technically feasible, but decided not to pursue that option because of the difficulty of the frequentist particle filtering inference framework to infer it jointly with the transmission parameters accounting for transmission dynamics stochasticity. The capacity to add priors on parameter values in the Bayesian statistical modeling framework enabled robust inference of the time-varying sensitivity using established methods (Perez-Saez et al., 2021).

Regarding the comparison of outbreak attack rates, we have added a novel figure to the Supplementary Material which shows the alignment in the estimates (Figure D below, now Supplementary Figure S11). We note that we do not expect these estimates to be identical for multiple reasons, including that the dynamic model explicitly accounts for the unobserved state of class members (pupils and staff) that were not recruited in the study.

Figure D: Comparison of predicted positive serologies from the statistical and dynamical models. Results are grouped by outbreak, sampling date and school group. Dots indicate the mean, horizontal bars the 95% CrI of the posterior distribution based on 5000 HMC draws and vertical bars of the smoothing distribution based on 1000

particles, vertical bars the 95% CI of the smoothing distribution based on 1000 particles. Dotted lines indicate the 1:1 ratio representing perfect model fidelity to data.

[3.4] While I don't doubt that sustained external introductions from the community makes school-based interventions less impactful in schools, nor that context dependent assessment is a good idea, I'm not sure that the authors provide much evidence that school-based interventions could not provide control when "community" transmission is high. Indeed, the conclusion of the paper is questionable due to the definition and simulation of "community" incidence to represent school based interventions.

We thank this Reviewer for highlighting what seem to be intuitive results of our study, which however stand in contrast with evidence on the timing of school closures during the COVID-19 pandemic (see response to comment [1.9]). We believe that this discrepancy supports the public health value of our results.

Regarding our choices on the scenario modeling study, we provided additional information in the new Methods section (see answer to comment [1.6]). It was unfortunately unclear to us what this Reviewer intended by “*community*” incidence to **represent** school based interventions”. We here clarify that in our previous scenario simulations community-acquired transmission rates were held constant across intervention scenarios (degrees of within-class and between-class intervention severity), and varied across three scenarios of epidemiologic contexts based on the three SARS-CoV-2 VOC periods in our study. As detailed below, we now provide additional results regarding the potential feedback effect of school-based interventions on community transmission. Using published estimates of school-closure impact on SARS-CoV-2 transmission during the COVID-19 pandemic, we conclude that our main results remain valid after accounting for this feedback effect.

[3.4] Here, the authors perform simulations with community incidence unaffected by school-based interventions to illustrate their conclusion in Fig 4C. However, one may wonder how much sustained activity would remain in the community if school-based interventions were generalized. In the data presented, "community" incidence reflects, among others, epidemic activity in the schools not included in this study. As incidence in schools is much higher than in the community, it would presumably change if school-based interventions were generalized, in a proportion linked to the contribution of children to transmission - disease and wave dependent. Consequently, the simulations basing the "counterintuitive" main result may correspond to the very situation that would be avoided with the adoption of school based interventions. Only a whole population model could provide more insight into that (and this has been done a lot during the COVID-19 pandemic).

We thank this Reviewer for raising this important point. In our previous scenario modeling simulations we had indeed held constant the community-acquired transmission rate in each of the three epidemiological setting scenarios, thus assuming that school-based interventions did not have a feedback effect on community-level transmission.

We have now performed novel simulation analyses when considering the feedback effect of school-based NPIs on community-wide transmission. These novel simulations are now described in the Methods section:

“As interventions in schools may have a feedback on the reduction of community-wide transmission, we perform additional simulations considering a range of reductions in community-acquired infections for each transmission setting. Empirical estimates suggest that school closure, the most stringent intervention scenario, during the COVID-19 pandemic reduced community-wide

transmission ranged between no effect and 60% , with most estimates of effect ranging between 25% and 50%. For each intervention scenario, we consider two community-transmission scenarios in addition to the baseline scenario (our inferred parameters), linking the reduction in the rate of community-acquired infections to the intensity of school-based interventions: one where the reduction in community-wide transmission increases linearly from 0 (no interventions) to 50% at school closure. The reduction for each intervention scenario is computed assuming that within-class and between-class reductions have an equivalent effect on the reduction of community-wide transmission (see Supplementary Material Section S1.9.2 for details).”

We present the results of these new simulations in Supplementary Figures S12-S14 As shown in Figure E below (now Supplementary Figure S13) The main outcome of these simulations is that our conclusion holds in the presence of feedback between reduction in within-school transmission and reduction in community transmission, ie. that at too high community transmission levels, school-based NPIs may be inefficient in reducing infections in the school-aged population.

Figure E: Sensitivity analysis of scenario simulations along the diagonal of scenario space by whether feedback was accounted for or not. Legend as in Figure 4b of the main text.

We now mention the robustness of our results in the Results section:

“These results were robust to different durations of epidemic simulations (6 and 12 months) (Supplementary Figure S12-S13), and when accounting for feedback between reductions in within-school and community-wide transmission (Supplementary Figures S12, S14).”

minor points :

M1 - "Further, SARS-CoV-V-2 RT-PCR testing was encouraged..." should it be SARS-CoV-2 ?

Thank you, this has been corrected.

M2 - On Figure 1, the empty dots are barely visible, and their colour hard to see. Full dots

would certainly be better.

Thank you for the suggestion, we have updated the figure accordingly.

M3 - SI - S9. (ref NBA, Alpha paper in England) please provide correct reference

The references have been updated.

M4 - S1.8.1 how were dates of epidemic defined ? Also a table with the t_L , t_R could be provided

Our study period was defined by the times of the first SeroCoV-Schools baseline survey (March 2021) to the last outbreak serological survey (February 2022). The dates to define VOC periods were chosen based on estimated VOC prevalence using available sequence information in the canton of Geneva (Figure 1).

M5 - S1.8.1.2.1,21,3: There is some confusion between implementation and model here. As a result, it makes it difficult to see whether the model is correct. The authors should try to present the probabilistic model rather than STAN tricks they used to implement it.

We hope that the new Methods section describing the statistical model now provides a clearer overview of our approach, and we are happy to update these sections should there still be points that this Reviewer would like to see clarified.

The model could be something like $P(Y_{it}, Z_{it_test}, t_{iinf})$ where Y_{it} is the serology result and Z_{it_test} the result of the RADT. that you split as $P(Y_{it}| t_{iinf}=t_{test}, Z_{it_test}) P(Z_{it_test}| t_{iinf}) P(t_{iinf})$

We have updated paragraph S1.8.1.2 to better clarify the observational model and likelihood function, focusing on what are data and what are unobservables we marginalize over.

.1 $P(Y_{it} | Z_{it}=1, T_{iinf})$ First, what is this "multinomial" for y^* ? in a multinomial, the sum of probabilities should be 1, it is not the case for p_1 and p_2 .

You mention t_{test} and t_{iinf} , but we don't know what is the relation between them : is the first taken to be a proxy of the second ?

We thank this reviewer for noting this point. These probabilities can be expressed as the conditional probabilities of a positive serology at time t conditional on a positive RT-PCR/antigen test at time t_{iinf} in the past ($t_{iinf} < t$). In writing down the equations for p^* we had omitted the normalization constant of the positive test result. We have corrected this, and the equations now read:

$$\begin{aligned} \mathbf{y}_{i,t}^* &\sim \text{multinomial}([p_1^*, p_2^*]), \\ \mathbf{y}_{i,t}^* &= \begin{cases} [0, 1] & \text{if } y_{i,t} = 1 \\ [1, 0] & \text{if } y_{i,t} = 0 \end{cases} \\ p_1^* &= \frac{\theta_{sero}^+(\tau_{i,t})\theta^+\phi_i(t) + (1 - \theta_{sero}^-)(1 - \theta^-)(1 - \phi_i(t))}{\phi(t)\theta^+ + (1 - \phi(t))(1 - \theta^-)}, \\ p_2^* &= \frac{(1 - \theta_{sero}^+(\tau_{i,t}))\theta^+\phi_i(t) + \theta_{sero}^-(1 - \theta^-)(1 - \phi_i(t))}{\phi(t)\theta^+ + (1 - \phi(t))(1 - \theta^-)}, \end{aligned}$$

We note that this error in notations did not have an impact on our results as the multinomial probability mass function in Stan automatically normalizes the vector that is passed to it to compute the individual outcome probabilities.

What happens if there are several t_{test} for the same person, and then how would it affect the computation?

If we have several positive tests for the same person we take the date of most distal infection, as we expect sensitivity to vary the most at short times post-exposure and only decrease slightly with time post infection within the time windows allowed by our study period (order of 1 year).

When you mean earliest here, is it the earliest in the follow-up or the last one before the serology ?

It means the earliest one before serology. We have clarified this in the Supplementary Material Section S1.8.1.

.2: same as before, what is the notation $\text{Bernoulli}()$? Shouldn't this be $P(y=1) = \phi(t) \sum_j (\theta^{t-t_j} P(t_j)) + (1-\phi(t)) (1-\theta)^{t-t_j}$

$\text{Bernoulli}()$ here denotes the probability mass function of the Bernoulli distribution. Yes we agree that for $P(y=1)$ this is correct. We here wanted to concisely write the likelihood for the general case ($y=1$ or $y=0$), therefore opted for the notation using the $\text{Bernoulli}()$ notation. We have now clarified the meaning of Bernoulli .

.3: This is $P(z_{it} | t_{iinf})$

M6 - S2.5 - the model does not seem to match serology at baseline. Why is it so and what are the consequences of that?

We believe that this mismatch may be due to the model relying on the population-representative seroprevalence data at baseline which seem to suggest lower seroprevalence than the available seropositivity data at baseline. A possible impact of this under-estimation is that the model slightly overestimates community-acquired infections in the initial phase of the study period (Alpha-dominated) to match available serology data at baseline in the 2021-2022 school year. However, given the high fidelity of the statistical model in reproducing outbreak sizes and final seropositivity data we do not believe that this initial under-estimation has an impact that carries over to the other VOC periods.

References

Banholzer, N., Van Weenen, E., Lison, A., Cenedese, A., Seeliger, A., Kratzwald, B., ... & Vach, W. (2021). Estimating the effects of non-pharmaceutical interventions on the number of new infections with COVID-19 during the first epidemic wave. *PLoS one*, 16(6), e0252827.

Banholzer, N., Feuerriegel, S., & Vach, W. (2022). Estimating and explaining cross-country variation in the effectiveness of non-pharmaceutical interventions during COVID-19. *Scientific reports*, 12(1), 7526.

Giardina, J., Bilinski, A., Fitzpatrick, M. C., Kendall, E. A., Linas, B. P., Salomon, J., & Ciaranello, A. L. (2022). Model-estimated association between simulated US elementary school-related SARS-CoV-2 transmission, mitigation interventions, and vaccine coverage across local incidence levels. *JAMA Network Open*, 5(2), e2147827-e2147827.

Perez-Saez, J., Zaballa, M. E., Yerly, S., Andrey, D. O., Meyer, B., Eckerle, I., ... & Specchio-COVID19 Study Group. (2021). Persistence of anti-SARS-CoV-2 antibodies: immunoassay heterogeneity and implications for serosurveillance. *Clinical Microbiology and Infection*, 27(11), 1695-e7.

REVIEWER COMMENTS

Reviewer #1 (Remarks to the Author):

Thank you for your thorough responses to my comments from the initial round of review. I am happy to see the methods generously expanded and feel that the paper is now much easier to understand. I have some remaining points that I would like to raise.

Thank you for the positive appreciation of our revisions.

1. I still struggle a little with the framing of the conclusions. And from your responses I think you speak more clearly about the results. What you have measured is a reduction in the risk contribution of school-based infectious exposures as total prevalence increases. It seems to me that the discussion of different VOCs is truly just a proxy for periods of varying community prevalence. It might therefore be clearer to discuss your results as the former rather than starting with 'successive VOCs' as the chief explanatory component.

We agree that the main variable that drives results in our scenario simulations is the degree of community-level transmission, and thus the importance of probability of community-acquired infections relative to within-school transmission. This can in theory be the result of multiple drivers, ranging from pathogen characteristics to socio-behavioural factors, and our intention was not to stress one over the other. Rather, our main discussion point is that all these factors have the potential to change in time, for instance due to pathogen evolution or changes in policy and public awareness, resulting in shifts in transmission patterns, and that this dynamic nature should be accounted for in the way school-based control measures are discussed and planned.

This being said, we believe that in our study the main drivers of change were related to SARS-CoV-2 VOC biological characteristics with respect to other socio-behavioural factors. NPI measures were relatively stable during our study period both in the schools included in our study and at the national level, as mentioned in the first paragraph of the results and illustrated in Supplementary Figure S2 on changes in NPIs stringency indices. In addition, Google mobility data from the canton of Geneva suggests that the three VOC periods did not show major differences (Figure A), although an in-depth analysis of this data is beyond the scope of this work. Although we acknowledge that mobility data cannot capture the full depth socio-behavioral dynamics and other changes may have occurred, it is unclear to us whether these may be large enough to explain inferred increases community-acquired SARS-CoV-2 infections. In contrast, evidence on the biological changes in VOC transmissibility and immune evasion capacity is well established (Cao et al. 2021), including their natural history parameters which we account for in our analysis, as well as the association between VOC emergence and large shifts in epidemiologic patterns across countries and settings (Elliot et al. 2022).

Taken together, we believe these elements substantiate our choice to subdivide the study period into periods of distinct VOC dominance to differentiate periods of within-school and community-level SARS-CoV-2 transmission.

We have updated the manuscript to clarify the distinction between the choice to use VOC periods in our study, and the main discussion point on changes in transmission dynamics and their impact on school-based NPIs. All mentions of VOC “waves” were replaced with references to VOC “periods”, as now introduced in the first paragraph of the Results section:

*“We conducted reactive outbreak investigations between March 2021 and February 2022, covering three successive periods of dominant SARS-CoV-2 variants (Alpha: **March-June 2021**, Delta: **July-December 2021**, and Omicron BA.1/BA.2: **December 2021-February 2022**, Figure 1A), henceforth referred to as “VOC periods””.*

and have explicitly mentioned our choice of focusing on VOC periods in the Discussion with a reference to novel Supplementary Figure S16 on human mobility data:

“Although changes in inferred transmission parameters cannot fully be attributed to VOCs alone, we note that during the study period NPI stringency in Switzerland was relatively stable (Supplementary Figure S2), and that measures of human mobility did not show strong variations (Supplementary Figure S16). In contrast, the emergence of the Alpha, Delta and Omicron subvariants has been consistently associated to strong increases in incidence globally, in particular for Omicron, thus highlighting the intrinsic potential of pathogen variants to shape epidemiologic patterns and motivating the focus in this work on the distinct periods of shifting VOC dominance.”

In addition, we have added references to the potential importance of socio-behavioural factors in driving changes in community-level transmission in the Results and Discussion sections:

*“We took the Alpha, Delta and Omicron periods as exemplars of epidemiological contexts along a gradient of increasing **transmission potential** and community infectious pressure, **noting that these contexts are driven both by difference in pathogen characteristics as well as contextual socio-behavioural factors.**”*

*“Taken together, these conclusions highlight the importance of continuous epidemiological surveillance to track community-level infection rates, the emergence of novel pathogen variants, **and changes in socio-behavioral factors favoring transmission when assessing the adequacy of different NPIs.**”*

“In addition, enhanced epidemiologic surveillance may also benefit from an integrated assessment of socio-behavioral and policy factors that have the potential to impact community-level transmission patterns, as exemplified by the strong changes in human mobility patterns during the COVID-19 pandemic.”

Figure A: Google mobility data for Geneva, Switzerland. Novel Supplementary Figure S16 showing changes in human mobility patterns based on publicly-available google mobility data during the study period.

2. I also want to come back to the point about generality. I disagree with the point about total prevalence always reducing the impact of school closures. In the instance where transmission is much higher amongst school children this would not necessarily be the case, or at least the relevance of this would be substantially reduced, for example in the context of an influenza pandemic where age-dependent immunity profiles play an important part in transmission dynamics (e.g. 2009). I recommend that this should be addressed directly in the discussion.

Thank you for this point, which we believe is an important one. We agree that our conclusion does not generalize to circumstances where the community does not contribute meaningfully to pathogen transmission to the schooled population. This could be due to very strong age-specific immunity profiles as in the case of highly immunizing diseases such as measles in a mostly vaccinated populations, or possibly like in the case of the 2009 influenza pandemic as mentioned by the Reviewer. However, we note that the main dimension that characterizes our different simulation scenarios is not exactly disease prevalence in the community, but rather the strength of community-acquired infections in the school population, which can correlate with prevalence but also incorporates susceptibility/infectivity profiles and age-mixing patterns. As such, it seems unlikely that in the presence of strong age-specific profiles of susceptibility or infectivity in the population the level of community-acquired infections in schools would be a driving factor of within-school transmission dynamics and infection patterns, at least not to the degree we have shown in our study.

In other words, we agree that in the theoretical case where community prevalence is large but community-acquired infections in school-aged children are low, school-based NPIs should be efficient in reducing incidence rates as the main driver of infections in schools will be within-school transmission. This statement is in agreement with our simulation results. It is however unclear whether this is a situation that has practical policy relevance, as this would require a combination of factors that can decouple high community disease prevalence from community-acquired infections in school-aged populations. We further note that cumulative attack rates during the 2009 influenza pandemic were much lower than the ones observed during the Omicron period across all age groups (Miller et al., 2010), and therefore believe that this example does not necessarily meet the conditions of high community-acquired infection rates in school-aged populations suggested in our scenario simulations. As such, we believe that our results align with this Reviewer's comment in that they do not suggest that school-based NPIs would have failed in reducing infections in the school-aged population in this specific setting.

Another dimension this Reviewer comment may refer to is the way age-specific profiles in susceptibility/infectivity may modulate the impact reduction of within-school transmission has on community-level transmission. In our previous round of reviews we have added scenario simulations that account for a feedback effect of school-based NPIs in reducing the level of community-acquired infections, assuming that school closures lead to a 50% reduction in community-level transmission. This is in line with published estimates during the COVID-19 as mentioned in the Methods section, but also for the 2009 influenza pandemic (Earn et al. 2012). We found that this additional feedback mechanism did not undermine our main conclusion on the qualitative reduction in school-based intervention effectiveness when community-acquired infection probability is very high. In this perspective, we believe our additional simulation analyses in the previous round of reviews addresses this point.

In light of these points, we have nuanced our summary statement in the Discussion section to better reflect our simulation studies:

*“These simulations demonstrate that school-based interventions can be highly effective when community transmission is low **and thus community-acquired infections by school-aged children rare, but can be largely ineffective when community transmission rates are high.**”*

We have also added elements to the limitations paragraph, adding the example of age-specific immunity profiles for the 2009 influenza pandemic, nuancing the general applicability of our results, and stressing the importance of future evaluation of our main conclusion to age-specific differences:

*“Finally, the results and conclusions of our scenario simulation investigations on the use of school-based measures in varying transmission settings are anchored in our study of SARS-CoV-2, and the same type of comparison may yield different quantitative results using natural history parameters and age-specific transmission rates for other respiratory pathogens, **for instance due to age-specific immunity profiles exemplified by the 2009 influenza pandemic.** [...] We nevertheless expect that our main result, ie. that school-based interventions loose effectiveness when **the probability of community-acquired infection is too high, is of policy relevance across a wide space of respiratory pathogen characteristics and may be broadly relevant** to infectious diseases involving school-aged children. The clear definition of what “too high” remains subject of future work for specific pathogens and transmission settings of public health importance, **in particular to assess the sensitivity of losses in school-based NPI effectiveness to age-specific profiles in pathogen exposure, susceptibility and infectivity.**”*

3. Finally, reading the paper from the start, I still find it hard to glean precisely how each result was acquired, i.e. without skipping to the methods. I think a sentence to put each result in context (methodologically) would support understanding here.

We have added contextual sentences at the beginning of the Results paragraphs:

“This statistical model accounted for changes in the underlying individual-level probability of SARS-CoV-2 infection their due to community-acquired infections or to within-school outbreaks which could vary between VOC periods and specific outbreaks.”

*“**We gained additional insight into viral introduction frequency through** phylogenetic analysis of 61 high quality school-based SARS-CoV-2 sequences from 7 of the 11 outbreaks **covering all three VOC periods, supplemented by a large set of community sequences from Geneva and Switzerland** (Supplementary Material section S1.5, Supplementary Figure S4).”*

“This dynamic model consisted of an SEIR-type model that simulates SARS-CoV-2 transmission events between individuals in schools, accounting for contact networks between students and staff (within- vs. between-group transmission), VOC-specific natural history parameters (incubation period and generation time) and levels of cross-variant immunity following infection or vaccination.”

Reviewer #2 (Remarks to the Author):

I am not entirely satisfied with how my comments have been incorporated into the revised version. My core argument was that school-based interventions should be understood as part of a broader package of measures aimed at reducing transmission across the entire population. The ultimate goal of such interventions is population-level protection. From the outset, the rationale behind these non-pharmaceutical interventions (NPIs) has been their contribution to lowering the effective reproduction number (R). Specifically, for school-based interventions, this has always been about assessing how schools contribute to the effective R. If their contribution is small, the interventions within schools will only have a limited impact, as transmission is largely driven by factors outside of schools. This point has been clear from the beginning. Hence, I feel the authors still need to better articulate how their modelling and results contribute new insights within this established framework. As it stands, the text still creates confusion for me. It still appears to me that the authors first isolate school-based interventions from their overarching objective and focus narrowly on their ability to mitigate the burden within schools. However, this reasoning is for me still missing the point. As a result, I would like to request that the authors revise their discussion to a greater extend

and reflect more explicitly what new insights their work provides. Specifically, it could highlight the lessons learned about estimating the contribution of schools to the overall effective R . For example, how might we define when the transmission originating from schools is “too low” warranting an exclusion of school based NPIs to protect the overall population? Additionally, the authors could propose frameworks, methodological approaches etc.

Thank you for these clarifications. We first wish to note that the general motivation and framing of our study aligns with this Review’s comment on the importance of schools as potential targets for NPIs due to their role in fueling community-wide epidemics. As such, we had stressed in the Introduction and Discussion that school-based interventions need to balance negative impact on pupils with epidemic control objectives. The premise of our work is that to better address this issue, it is key to understand within-school transmission dynamics to assess the eventual effect of NPIs in schools. In fact, any modeling study assessing the role of school-based NPIs on epidemic progression will need to make assumptions on how they will change infection patterns in school-aged populations and their contribution to community-level transmission through contact networks.

In this perspective, we believe our work provides novel insight by showing the degree to which within-school pathogen transmission dynamics can change in time as a function of the level of community-acquired infections, here exemplified with the COVID-19 pandemic. As suggested by this Reviewer, the impact of school-based interventions on community-level epidemic outcomes will depend on the level of reduction of infections in the school-aged population through the number of secondary infections they carry on to the rest of the community. We here chose to keep the focus on the reduction of infections in the school-aged population as we believe that this makes clearly the point that if infection rates in this population do not change due to school-based NPIs it means that these NPIs are failing in their necessary condition to have wider community-wide impacts, as the number of secondary infections due to school-aged individuals will remain unchanged. Our inferential modeling shows that transmission dynamics can change drastically in time, and as a consequence the effectiveness of school-based NPIs can also change, up to including failure when community-level transmission is particularly strong.

We acknowledge that we did not implement full community-wide transmission models that incorporate mixing patterns to explore quantitatively the impact of school-based NPIs on community-level epidemic outcomes and effective reproductive numbers. We however believe that this is out of the scope of our work as we did not aim to quantify the exact contribution of school-based NPIs to R_{eff} in the specific setting of the COVID-19 pandemic, nor in specific scenarios. Our aim was rather to provide a qualitative view on the potential impacts of changing levels of community transmission on school-based NPIs, and we believe that focusing on infection outcomes in the school-aged population was sufficient to this reach this aim. Furthermore, in reply to the previous round of Reviews we accounted for feedback effects between school-based NPIs and the probability of community-acquired infections, showing that our qualitative result hold, which we believe further strengthens the conclusions of our study. Our main conclusion is equally a qualitative one, ie. a call for a dynamic view on the role of school-based NPIs in epidemic control as a function of changing transmission dynamics, and believe that this has policy relevance across a wide range of settings and pathogens (see reply to point 2 of Reviewer 1 above). As such, we do not believe we have specific lessons learned to share about the contribution of schools to overall effective R , which we did not undergo in our inferential modeling analyses nor considered in the scenario simulations, but rather wished to state the importance of taking changing transmission dynamics into account when assessing future policy decisions regarding school-based NPIs.

We have now stated the value of our work more clearly in the first paragraph of the Discussion:

“Through an integrated analysis of high-resolution epidemiologic and virological data this work reveals the degree to which within-school pathogen transmission dynamics can change in time as a function of the level of community-acquired infections, here exemplified with the COVID-19 pandemic, and its potential impact on school-based NPI effectiveness.”

We have also clarified the aim of our scenario analysis and justification to focus on school-aged infections as main outcome in the Discussion:

“We note that our simulations focused on infection rates in the school-aged population, and not in resulting community-level changes in transmission intensity which would of required a full simulation of community-level transmission dynamics. Because our aim was to provide a qualitative view on the potential impacts of changing levels of community transmission on school-based NPIs, we considered reduction in school-aged population infections a sufficient proxy of population-level impact as it maps to subsequent reduction in secondary transmission events, echoing previous studies with similar aims.”

Taking queue for this Reviewer’s comment, we have added a sentence on the integration of surveillance systems modeling efforts to assess the potential effectiveness of school-based NPIs:

“Identifying these situations will benefit from continued efforts in real-time inferential modeling of changes in epidemic progression, as well as in strategic modeling to compare alternative scenarios of epidemic drivers and public health interventions.”

Reviewer #3 (Remarks to the Author):

I thank the authors for the added work.
I have no further comment.

We thank this reviewer for the positive appreciation of our work.

References

Cao, Y., Wang, J., Jian, F., Xiao, T., Song, W., Yisimayi, A., Huang, W., Li, Q., Wang, P., An, R. and Wang, J., 2022. Omicron escapes the majority of existing SARS-CoV-2 neutralizing antibodies. *Nature*, 602(7898), pp.657-663.

Earn, D.J., He, D., Loeb, M.B., Fonseca, K., Lee, B.E. and Dushoff, J., 2012. Effects of school closure on incidence of pandemic influenza in Alberta, Canada. *Annals of internal medicine*, 156(3), pp.173-181.

Elliott, Paul, Oliver Eales, Nicholas Steyn, David Tang, Barbara Bodinier, Haowei Wang, Joshua Elliott et al. "Twin peaks: the Omicron SARS-CoV-2 BA. 1 and BA. 2 epidemics in England." *Science* 376, no. 6600 (2022): eabq4411.

Miller, E., Hoschler, K., Hardelid, P., Stanford, E., Andrews, N. and Zambon, M., 2010. Incidence of 2009 pandemic influenza A H1N1 infection in England: a cross-sectional serological study. *The Lancet*, 375(9720), pp.1100-1108.

REVIEWER COMMENTS

Reviewer #1 (Remarks to the Author):

First, thanks for your thorough responses. Overall, I am happy with the changes and it is now much easier to follow..

Thank you for your positive feedback.

My only remaining request is that you tone down the sentence in the discussion:

"These simulations demonstrate that school-based interventions can be highly effective when community transmission is low and thus community-acquired infections by school-aged children rare, but can be largely ineffective when community transmission rates are high."

I think a more appropriate phrasing might be along the lines of:

"These simulations demonstrate that school-based interventions can be highly effective when community transmission is low and thus community-acquired infections by school-aged children rare. However, when community transmission rates are high, school-based interventions can only be implemented effectively as part of a comprehensive suite of interventions that target transmission in multiple settings."

Thank you for your suggestion, which we believe accurately summaries our findings. We updated the text accordingly.

Reviewer #2 (Remarks to the Author):

I thank the authors for their clarifications. But I am sorry, I respectfully note that I believe I dispute their premise.

They write: "The premise of our work is that to better address this issue, it is key to understand within-school transmission dynamics to assess the eventual effect of NPIs in schools. In fact, any modelling study assessing the role of school-based NPIs on epidemic progression will need to make assumptions on how they will change infection patterns in school-aged populations and their contribution to community-level transmission through contact networks."

While I appreciate the authors' perspective, I do not believe this is necessarily the case. Network-based transmission models can analyze the propagation of disease within a population, even without explicitly modelling the within-school dynamics. Such models can nonetheless draw conclusions about the potential contribution of schools to overall transmission and, by extension, about the possible impact of school-based NPIs.

For example, population-level transmission patterns would be markedly different depending on whether schools contribute significantly to transmission or not. The timing and spatial distribution of infections among adults, for instance, would differ between scenarios where schools act as a major transmission node versus scenarios where they do not. This distinction arises because transmission pathways originating in schools differ from those originating in other settings, such as workplaces.

However, I do agree with “their contribution to community-level transmission through contact networks”. You need to understand transmission patterns within families to make a quick assessment regarding the contribution of schools. When children often infect their parents, school based NPIs are a good idea, when it is the other way around – and parent bring infections to households – this is less likely to be the case.

But when I dispute their premise I don't believe I dispute their work – only the discussion. Hence, I don't think my point would justify to block this publication further. I just hope that the authors could be more explicit about the need to explicitly model within school dynamics to address this particular question of the impact of NPIs within a pandemic.

We appreciate the thorough appreciation of our work and commitment to clarifying the contextualization of this study. We believe that the previous rounds of review may not rightfully reflect our agreement with some of the points raised by this Reviewer, and will try to highlight the points which we would like to re-state in a clearer which may help reconcile the difference in the appreciation of our premise.

Firstly, and as stated in our previous rebuttal, we agree that the comprehensive assessment of the effect of school-based NPIs on epidemic progression may be valuably informed by transmission models that account for contacts between school-aged children and the rest of the population, starting from their households. We also agree with the statement that “Network-based transmission models can analyze the propagation of disease within a population, even without explicitly modelling the within-school dynamics.”.

However, the main point we would like to make is that when assessing the impact of school-based NPIs, an assumption will need to be made on how the given NPI, or package of NPIs, will alter the transmission potential of schools. Irrespective of the modeling approach, including networks models, this effect will need to be accounted for either directly if within-school transmission is modeled, or indirectly by enforcing how NPIs change the level to which schools contribute to onward transmission with respect to alternative intervention scenarios.

We believe that the value of our study is in showing that the functional relationship between school-based NPI stringency and resulting reduction in school-aged infections can vary strongly depending on the overall epidemiologic setting, in particular in the strength of community-acquired infections. As such, this insight can help inform other types of modeling approaches that study the community-level effect of school-based NPIs in stressing that school-based NPI effectiveness also depends on the transmission context. As noted earlier in our exchanges, this may seem like an intuitive result, but seems valuable in light of how school-based interventions were implemented during the COVID-19 pandemic.

To further clarify this points, we have added to the limitations paragraph in the Discussion:

“Future modeling studies aimed to, and tooled for, assessing the effect of school-based NPIs on community-wide epidemic patterns may leverage these insights in the assumptions on how NPI stringency is mapped to the effective contribution of schools to onward community transmission.”

We hope these additions clarify our position, and help address this Reviewer's difference in perspectives.